# MIRACLE: Towards Personalized Dialogue Generation with Latent-Space Multiple Personal Attribute Control

**Zhenyi Lu[1,2] , Wei Wei[1,2]\*, Xiaoye Qu[1,2],**
**XianLing Mao[3], Dangyang Chen[4], Jixiong Chen[5]**

[1] Cognitive Computing and Intelligent Information Processing (CCIIP) Laboratory,
School of Computer Science and Technology, Huazhong University of Science and Technology
[2] Joint Laboratory of HUST and Pingan Property & Casualty Research (HPL),
[3] Beijing Institute of Technology,
[4] Ping An Property & Casualty Insurance company of China, Ltd.,
[5] Brilliance Technology Co. Ltd.

luzhenyi529@gmail.com,{weiw,quxiaoye}@hust.edu.cn,maoxl@bit.edu.cn
chendangyang273@pingan.com.cn, chenjixiong@brilliance.com.cn

## Abstract

Personalized dialogue systems aim to endow the chatbot agent with more anthropomorphic traits for human-like interactions. Previous approaches have explored explicitly user profile modeling using text descriptions, implicit derivation of user embeddings, or utilizing handicraft prompts for ChatGPT-like models. However, textual personas are limited in describing multi-faceted attributes (*e.g., language style, inner character nuances*), implicit embedding suffers from personality sparsity, and handicraft prompts lack fine-grained and stable controllability. Hence, these approaches may struggle with complex personalized dialogue generation tasks that require generating controllable responses with multiple personal attributes. To this end, we propose MIRACLE, a novel personalized dialogue generation method through **M**ult**I**ple Pe**R**sonal **A**ttributes **C**ontrol within **L**atent-Space **E**nergy-based Models. Specifically, our approach first disentangles complex personality into multi-faceted attributes. Subsequently, we employ a conditional variational auto-encoder to align with the dense personalized responses within a latent joint attribute space. We have also tailored a dedicated energy function and customized the ordinary differential equations sampling method to offer flexible attribute composition and precise attribute control. Extensive experiments demonstrate that MIRACLE outperforms state-of-the-art models regarding both personality controllability and response generation quality. Our dataset and code are available at https://github.com/LZY-the-boys/MIRACLE

## 1 Introduction

Building a personalized and anthropomorphic chatbot is an essential goal in the field of dialogue sys-

---

\* Corresponding author.

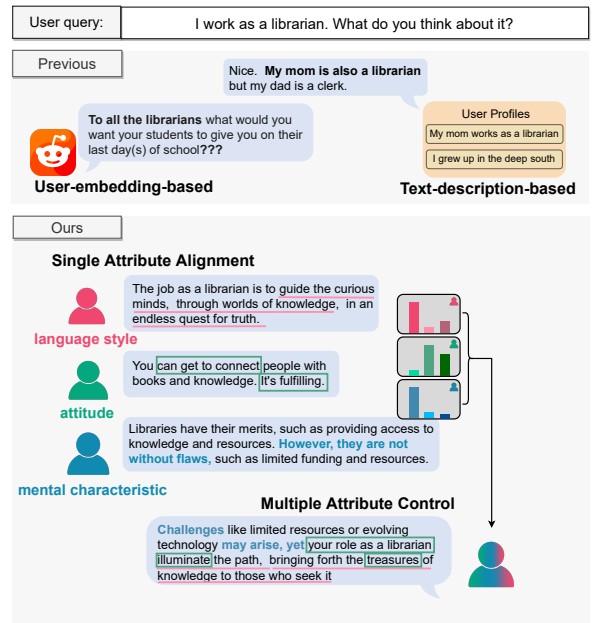

Figure 1: **Top:** previous methods model personas by user embedding derived from user posts (*e.g., in Reddit*) or a series of text descriptions. **Bottom:** Our approach models personality as the composition of multiple personal attributes. We train MIRACLE to align with different personal attributes (*language style, attitude, etc*), and control multiple attributes to represent diverse personalities during inference.

tems. It aims to endow chatbot agents with human-like traits, enabling more realistic interactions (Li et al., 2016b; Zhang et al., 2018; Wolf et al., 2019; Song et al., 2021a; Li et al., 2023). Studies in behavioral psychology reveal that humans have a natural tendency to attribute human-like traits to non-human entities (Qu et al., 2023; Gu et al., 2022, 2021) during interaction (Epley et al., 2007; Airenti, 2018). Therefore, personalization in dialogue systems has the potential to enhance user trust and enrich interaction experiences with Artificial Intelligence (AI) agents (Choung et al., 2022).

Recent personalized dialogue methods often rely on text descriptions (Song et al., 2019; Wolf et al., 2019; Xu et al., 2022; Chen et al., 2023) to model user profiles. However they primarily focus on concrete identifiable facts and background information, *e.g., age, job, location*, neglecting the multifaceted dimensions of personality (Moore et al., 2017; Ahn et al., 2023). For instance, while a statement like *"I grew up in the deep south"* conveys traits related to regional identity, it overlooks other personality dimensions such as language style, attitudes, and inner character nuances. Other methods for personalized dialogue generation often rely on user embeddings derived from social media platforms like Reddit (Qian et al., 2021; Ma et al., 2021; Huang et al., 2022; Zhong et al., 2022). However, these models encounter challenges due to the sparsity present in real-world posts, as they lack explicit persona modeling. Consequently, they may struggle to achieve accurate and comprehensive personalization through implicit embeddings.

While recent advancements in large language models, such as ChatGPT[1], have facilitated personalized content through manual prompts, it is non-trivial to directly impersonate a specific persona using such prompts (Zhuo et al., 2023; tse Huang et al., 2023). This challenge stems from the inherently ambiguous and limited expressiveness of prompts, failing to achieve precise control over personalized content.

In this paper, we present MIRACLE, a novel approach that enables more precise and reliable fine-grained control over personalization in dialogue systems. Specifically, we propose modeling user personality by disentangling it into multiple distinct personal attributes. As illustrated in Figure 1, personality can be decomposed into various attributes, including attitude, language style, mental characteristics, and more. Each attribute encompasses specific aspects, such as optimism or pessimistic for the attitude attribute. This decomposition allows us to capture the diverse dimensions of an individual's personality and enables fine-grained modeling and control of each attribute separately. By combining these aspects from multiple attributes, we can express a wide range of unique personalities. To achieve personalized generation, we specify an energy function that incorporates multiple personal attributes in a product-of-expert (POE) manner. By assigning lower energy to responses that better

[1] https://chat.openai.com/

align with the specified aspects, our approach enables personalized generation by sampling from an energy-based model (EBM), providing flexible and fine-grained control over the personalization of generated responses.

To address the challenge of personality sparsity and enhance personalized generation quality, we collect a high-quality multi-turn dialogue corpus, which is characterized by its dense coverage of each individual aspect. To circumvent the non-differentiable nature of the text and better align with the dense aspect data, we employ a conditional variational autoencoder (CVAE) framework (Sohn et al., 2015) to map the attributed dialogue to a shared latent space. To enhance attribute representation further, two new loss functions are introduced to promote the distinctiveness and compactness of the latent space. Within this latent space, we leverage the designed EBM to capture the aspect density and compose different attributes. Additionally, we utilize an adapted ODE sampling method to efficiently draw personalized responses from this distribution.

In summary, our contributions include a novel personalized dialogue generation approach through fine-grained control over multiple personal attributes in the CVAE-based latent space, with two new losses promoting distinct and compact attribute representations and flexible EBM-based composition of different personal attributes using a customized ODE sampling method. Experimental results demonstrate that our approach achieves state-of-the-art performance, striking a superior balance between generation quality and personalized control. A high-quality personal attributed dialogue corpus for research purposes is also provided.

## 2 Related Work

### 2.1 Personalized Response Generation

Existing methods for personalized dialogue generation can be broadly classified into two groups: text-description-based methods and user-embedding-based methods.

In the category of text-description-based methods, early works (Wolf et al., 2019; Song et al., 2020, 2021a) primarily focus on promoting persona consistency through pre-trained language models, while recent advancements borrow knowledge-enhance techniques (Liu et al., 2022b; Fu et al., 2022; Jang et al., 2022) and incorporate entailment/discourse relations (Chen et al., 2023). However,

these methods often represent personas as key-value lists or sentences, which limits accurately understanding and expressing personality nuances.

As for embedding-based methods, traditional approaches (Li et al., 2016b; Al-Rfou et al., 2016) attempt to exploit user ID information, while DHAP (Ma et al., 2021) embed user dialogue history as implicit profiles. More recently, contrastive learning (Huang et al., 2022), refined retrieval (Zhong et al., 2022) and CVAE-based clustering (Tang et al., 2023) are explored to enhance the personalization performance. However, these approaches may still suffer from the personality scarcity of real-world posts without explicit modeling. Additionally, utilizing implicit embeddings to guide personalization effectively remains a significant challenge.

## 2.2 Energy-based Text Modeling

Recently, energy-based models (EBMs) have emerged as a flexible generative framework capable of handling diverse configurations (Khalifa et al., 2021; Liu et al., 2022a). These models allow for the incorporation of arbitrary functions into the energy function, which is minimized during inference. As a result, many recent works leverage EBMs to model complex distributions (Pang and Wu, 2021; Yu et al., 2022) and incorporate multiple constraints and attributes (Nie et al., 2021; Pang and Wu, 2021; Qin et al., 2022; Liu et al., 2022a). For example, Mix-and-Match (Mireshghallah et al., 2022) employs EBMs to combine arbitrary black-box scorers for guiding text generation, while COLD (Qin et al., 2022) utilizes the energy function to impose arbitrary constraints during the decoding process. LatentOps (Liu et al., 2022a) introduces composable text control operations utilizing classifier-based EBMs. However, these works primarily focus on plain-text generation domains, whereas our approach applies EBM to dialogue-generation scenarios, specifically modeling complex personality as a composition of multiple personal attributes based on CVAE architecture. We also adapt the ODE sampling method to effectively sample personalized dialogue responses.

## 3 Methodology

### 3.1 Notation

**Task Definition**  The task is to generate a personalized response, denoted as $r_M$, given the personality $P$ and a multi-turn dialogue context $C =$ $\{q_1, r_1, \ldots, q_{M-1}, r_{M-1}, q_M\}$. Here, $q$ and $r$ represent the user query and chatbot response, respectively. In essence, the objective of personalized response generation is to estimate the probability distribution $p(r|C, P)$ in order to generate specific personalized responses.

**Personality Modeling**  In contrast to previous work, we propose a new approach to disentangle the personality $P$ as the composition of different persona-related attributes, represented by $\mathbf{P} = (P_1, P_2, P_3, \ldots, P_N)$, where $N$ is an arbitrary number and is easily adjustable. Each attribute $P_i$ may has $n_i$ candidate aspects, denoted as $P_i \in \{p_i^1, p_i^2, \ldots, p_i^{n_i}\}$.

Given a particular personality configuration $\mathbf{P} = (p_1^{a_1}, p_2^{a_2}, \cdots, p_N^{a_N})$, the objective of personalized response generation is to generate a response $r$ that incorporates these aspects simultaneously.

### 3.2 Single-Aspect Dialogue Data Collection

To ensure the alignment with dense attributes disentangled from personality, we curated a multi-turn conversation corpus for each specific aspect of these attributes. Leveraging the capabilities of ChatGPT in generating single-attribute data (Coda-Forno et al., 2023) and multi-turn conversations (Xu et al., 2023), we designed instruction templates to prompt ChatGPT to simulate two-person conversations. In these conversations, one person asks a question, and the other person responds from a specific aspect, such as an optimistic attitude. To enhance corpus diversity, we also pre-select a series of "seed" topics[2], around which conversations should be centered. To improve the aspect density of the collected corpus, we conducted multiple rounds of human evaluation and cleaning, resulting in a clean version of approximately 44k dialogue turns, further details of this process can be found in Appendix A. It is important to note that we collect single-aspect conversations for the training dataset, the multiple-attribute data is only collected for testing purposes due to its time-consuming nature caused by extensive combinations of different attributes[3].

---

[2]To ensure fair evaluation, we use persona descriptions from the PersonaChat (Zhang et al., 2018) as conversation topics (see Section 4.1).

[3]For example, if we consider three attributes, each with two aspects, there would be a total of eight combinations of these attributes.

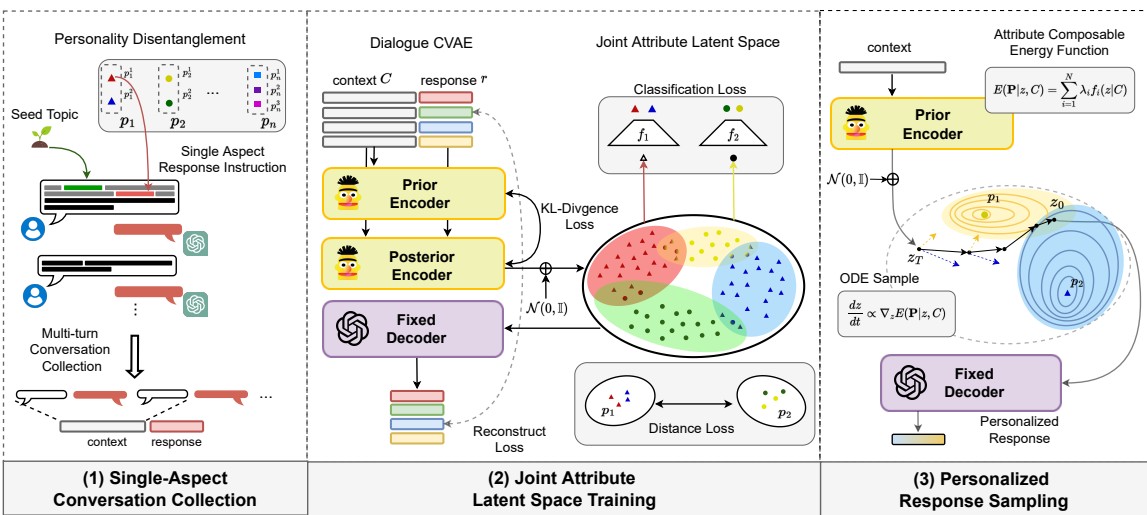

Figure 2: The overview of our MIRACLE method. **(1)** We collect a high-quality single-aspect conversation training corpus using the ChatGPT API (Section 3.2). **(2)** We construct a joint attribute latent space through a dialogue CVAE, and introduce the aspect classification loss and the attribute distance loss to enhance the distinctiveness and compactness of the attribute space (Section 3.3). **(3)** We design an energy function to compose each aspect within the joint latent space and draw desired vectors by ODEs sampling, which are then decoded to generate personalized response sequences (Section 3.4).

## 3.3 Joint Attribute Space Training

To facilitate the generation of personality-dense responses, we adopt a CVAE framework to map the aspect-specific dialogue data into a joint attribute space so that samples from the specific aspect space are aligned with aspect-dense response sequences. To further enhance this joint attribute space, we introduce two specific losses. The first loss focuses on promoting the distinctness of each aspect, while the second loss aims to increase the intersection between different attributes, allowing for fine-grained sampling over multiple attributes.

**Building CVAE** To construct the dialogue Conditional Variational Autoencoder (CVAE), we employ two distinct models as encoders: a posterior encoder $p_\theta(z|C, r)$ and a prior encoder $p_{\theta'}(z|C)$. Both encoders, based on the pre-trained BERT (Devlin et al., 2019), allow CVAE to effectively capture the given input context $C$ by latent variable $z$. During training, CVAE utilizes the posterior distribution to generate high-quality responses $r$, while during inference, when the response $r$ is unseen, the prior distribution is used to sample the latent variable $z$. Moreover, the GPT2 model (Radford et al., 2019) is leveraged as the decoder $p_\phi(r|C, z)$, where $\theta, \theta'$ and $\phi$ represent the trainable parameters of the posterior encoder, prior encoder, and decoder respectively.

Under the assumption that CVAE posterior and prior distribution follows an isotropic multivariate

Gaussian distribution, we compute the mean $\mu, \mu'$ and variance $\sigma^2, \sigma'^2$ by the two encoders:

$$
\begin{aligned}
h &= \text{Pooling}(\text{BERT}_\theta([C; r])) \\
h' &= \text{Pooling}(\text{BERT}_{\theta'}([C])) \\
\begin{bmatrix} \mu \\ \log \sigma^2 \end{bmatrix} &= \text{MLP}(h) \\
\begin{bmatrix} \mu' \\ \log \sigma'^2 \end{bmatrix} &= \text{MLP}'(h')
\end{aligned}
\tag{1}
$$

Subsequently, we utilize reparameterization technique (Kingma and Welling, 2013) to sample posterior $z$ and prior $z'$ from $\mathcal{N}(\mu, \sigma^2\mathbb{I})$ and $\mathcal{N}(\mu', \sigma'^2\mathbb{I})$. This technique enables a differentiable sampling process.

$$
\begin{aligned}
z &= \mu + \sigma\xi, \ \xi \sim \mathcal{N}(0, \mathbb{I}) \\
z' &= \mu' + \sigma'\xi', \ \xi' \sim \mathcal{N}(0, \mathbb{I})
\end{aligned}
\tag{2}
$$

Finally, the sampled latent variable $z$ (during training) or $z'$ (during inference) is fed into the GPT2 decoder to map it back to text space, resulting in the generation of a response.

CVAE is trained using stochastic gradient variational bayes (SGVB) (Kingma and Welling, 2013), which maximizes evidence lower bound objective (ELBO) of conditional log-likelihood. The ELBO consists of two components: a dialogue response reconstruction term that ensures the generative quality of the posterior distribution $p_\theta(z|C, r)$, and a regularization term that aligns the prior distribution $p_{\theta'}(z|r)$ with the posterior $p_\theta(z|C, r)$. This alignment fosters consistency during inference, where

the unseen response $r$ is generated.

$$\text{ELBO} = \underbrace{\mathbb{E}_{p_\theta(z|C,r)}[\log p_\phi(r|C,z)]}_{\text{Response Reconstruction gain}}$$
$$- \underbrace{\text{KL}(p_{\theta'}(z|C)||p_\theta(z|C,r))}_{\text{Regularization on } z} \quad (3)$$
$$\mathcal{L}_{\text{VAE}} = -\text{ELBO}$$

**Optimizing Joint Attribute Space** We introduce the aspect classification loss and the attribute distance loss. The **aspect classification loss** aims to improve the discriminability of latent representations for aspects within the same personal attribute. Specifically, we incorporate individual classifier heads for each attribute and train them using the cross-entropy loss:

$$\mathcal{L}_C = -\sum_{i=1}^{N}\sum_{j=1}^{|P_i|} y_{p_j}^{(i)} \log(\hat{\mathbf{y}}_{p_j}^{(i)}) \quad (4)$$

where $y_{p_j}^{(i)}$ represents the ground truth probability for class $p_j$ within the attribute $P_i$, and $\hat{\mathbf{y}}_{p_j}^{(i)}$ represents the predicted probability. By optimizing this aspect classification loss, we encourage the aspect representations to be more distinguishable, enabling more fine-grained sampling. An illustration of this concept can be found in the middle part of Figure 2 (*e.g.,* the red and blue aspect distribution of $P_1$ attribute exhibit clear separation).

Meanwhile, to encourage the model to capture intersections between different attributes, enabling the sampling of responses with multiple attributes simultaneously, we introduce an **attribute distance loss**. This loss penalizes the Euclidean distance between every two distinct attribute distributions. To avoid expensive computation, we approximate this loss on a batch level, taking the average within each mini-batch of size $B$:

$$\mathcal{L}_D = \sum_{1 \le a < b \le N} ||\frac{1}{B}\sum_{i=1}^{B} z_i^{P_a} - \frac{1}{B}\sum_{j=1}^{B} z_j^{P_b}|| \quad (5)$$

Minimizing such loss allows the model to reduce the conflicts between different attributes. (*e.g.,* $P_1$ and $P_2$ attribute has intersection in Figure 2)

To sum up, our final training objective is:

$$\mathcal{L} = \mathcal{L}_{\text{VAE}} + \mathcal{L}_C + \mathcal{L}_D \quad (6)$$

### 3.4 Personalized Response Sampling

We formulate personalized response generation as sampling response samples that contain multiple specific aspects of personality attributes.

To achieve fine-grained control over different attributes, we define an attribute-composable energy function that calculates the aspect density in the latent space. By leveraging adapted ODE sampling methods, we can efficiently draw samples of interest from this distribution.

**Latent EBM Formulation** In order to sample aspect-abundant vectors $z$ in the latent space, we utilize attribute-specific classifiers[4] denoted as $f_i$ to quantify the density of aspect $p_i^j$ from $z$, represented as $f_i(z)[j]$.

We utilize EBM to estimate the richness of personality expressed in the responses ($Z$ is the normalizing factor):

$$p(\mathbf{P}|z,C) = \frac{exp(-E(\mathbf{P}|z,C))}{Z} \quad (7)$$

where its energy function is designed in the POE manner to aggregate multiple personal attributes into a comprehensive representation of the overall personality (Outlined in Appendix B.1).

$$E(\mathbf{P}|z,C) = \sum_{i=1}^{N} E_i(P_i|z,C)$$
$$= \sum_{i=1}^{N} \lambda_i f_i(z|C)[a_i] \quad (8)$$

In this context, $\lambda_i \ge 0$ is the weight of $P_i$ attribute and $a_i$ is the desired aspect index of $P_i$.

The energy function $E(\mathbf{P}|z,C)$ can be interpreted as a linear combination of the richness of personal attributes. Thus sampling from this EBM with low energy corresponds to response sequences exhibiting a higher density of multiple selected aspects $p_i^{a_i}, i \in \{0, \cdots, N\}$. It is worth noting that we utilize this energy-based formulation only during the inference procedure, enabling arbitrary combinations of personal attributes without the need for combination-specific fine-tuning.

**ODE Personalized Sampling** Due to the intractable normalization factor $Z$, a common practice is to sample from EBMs rather than directly calculate it. In our approach, we derive the ODE sampling method based on CVAE to sample from such EBM. Specifically, in Appendix B.2, we demonstrate that the ODE in our CVAE latent space takes the following form:

$$\frac{dz}{dt} = \frac{1}{2}\beta(t)\left[\nabla_z \sum_{i=1}^{N} \lambda_i f_i(z|C)[a_i]\right] \quad (9)$$

---

[4]Those classifiers are trained by Equation 4

Here, the ODE is solved with negative time increments from $T$ to 0. To generate a sample $r$ that aligns with a specific personality $P$, the process involves drawing $z(T) \sim \mathcal{N}(z|C)$ and solving for $z(0)$ in the aforementioned equation using a black-box ODE solver[5] (Chen et al., 2018, 2021). Subsequently, the obtained $z(0)$ is decoded back to the text space to yield a personalized response.

Intuitively, in the right term of Equation 9, a higher value of $f_i(z|C)[a_i]$ indicates that the $z$ better aligns with the aspect $p_i^{a_i}$. By letting $\frac{dz}{dt} \propto \nabla_z f_i(z|C)[a_i]$, we can pull $z$ towards more aspect-abundant places that yield more personalized responses. The summation ensures that each aspect is taken into account so that we can incorporate multiple selected aspects in one sample.

## 4 Experiments

To verify the effectiveness of our proposed MIRACLE, we conduct extensive experiments on both automatic and human evaluations. Additionally, we provide further analysis on ablation, efficiency, and case studies.

### 4.1 Experimental Setups

**Dataset** To evaluate the personalization and generation capabilities of our approach, we focus on **language style** (with two aspect: lyrical/plain), **attitude** (optimistic/pessimistic), and **mental characteristics** (critical/emotional). We randomly sample 11,000 dialogue turns per aspect (a total of 132,000 utterances) from our collected multi-turn dialogue corpus for training our MIRACLE model. For evaluation, we use ChatGPT to generate conversations on different topics, covering eight combinations of the three personal attributes. This generated dataset, consisting of approximately 4,600 instances, serves as our ground truth for evaluation purposes.

**Baselines** For comparison, we select the following baselines: **(1) Text-description-based methods**: We compare with BOB (Song et al., 2021a) and LMEDR (Chen et al., 2023), both are strong text-description-based personalized models. **(2) User-embedding-based methods**: Our second set of baselines includes MSP (Zhong et al., 2022), and CLV (Tang et al., 2023). To ensure a fair comparison, we randomly select personas from the PersonaChat dataset (Zhang et al., 2018) as conversation topics when generating our data, and feed the

topics as personas input to BOB, CLV and LMEDR during training. More detail of the baseline can be found in Appendix C.1

### 4.2 Evaluation Metrics

In order to obtain accurate and comprehensive performance comparisons, we use both automatic and human evaluations.

**Automatic Evaluation Metrics** We assess the quality of dialogue responses from four perspectives: **(1) Personalization**: To evaluate the personalization of the generated responses, we employ attribute-based text classifiers to measure the accuracy score of each attribute in the generated responses (Mireshghallah et al., 2022). Additionally, we report the average score across the three attributes to assess the overall effect of personalization. **(2) Coherence**: Coherence is measured using BLEU and Rouge metrics at the word overlap level. We also utilize Natural Language Inference (NLI) to evaluate the semantic coherence, as suggested by previous work (Liu et al., 2022b). **(3) Fluency**: To assess the fluency of the generated responses, the negative log-likelihood of the generated responses according to the GPT2-XL[6] is used as the fluency score (Chen et al., 2023; Qin et al., 2022). **(4) Diversity**: We measure the diversity of the generated responses using the Distinct metrics and the self BLEU score (sBLEU) as proposed in (Tang et al., 2023; Liu et al., 2022a). Further details can be found in Appendix C.3.

**Human Evaluation Metrics** Consistent with prior studies (Tang et al., 2023; Chen et al., 2023), we conduct human evaluations on 100 randomly selected test samples. Three annotators assess the generated responses for readability, personalization, and coherence in a double-blind manner. We calculate the Fleiss Kappa value of 0.63, indicating substantial agreement among the annotators (Gwet, 2014). The evaluations are normalized into specific scores on a scale of [0, 1].

### 4.3 Experimental Results

**Automatic Evaluations** The performance of all models on different automatic metrics is presented in Table 1. Notably, our MIRACLE model demonstrates substantial improvements in personalization metrics while maintaining good generation quality. Specifically, the following observations can be

---

[5] https://github.com/rtqichen/torchdiffeq

[6] https://huggingface.co/gpt2-xl

| Methods | Personalization | | | | Coherence | | | Fluency | Diversity | |
|---|---|---|---|---|---|---|---|---|---|---|
| | Avg.↑ | L.↑ | A.↑ | M.↑ | BLEU↑ | Rouge↑ | NLI↑ | PPL↓ | Distinct↑ | sBLEU↓ |
| BOB | 58.54 | 54.18 | 65.24 | 56.19 | 39.52 | 13.47 | 60.56 | 55.18 | 64.04 | 15.00 |
| MSP | 62.83 | 65.70 | 66.04 | 56.75 | 36.36 | 11.88 | 40.89 | 78.87 | 66.55 | 11.02 |
| CLV | 59.91 | 55.50 | 67.73 | 56.50 | 37.13 | 13.26 | 68.33 | 44.12 | 64.50 | 15.56 |
| LMEDR | 72.67 | 78.96 | 79.06 | 60.00 | 44.50 | **16.50** | 72.23 | 21.78 | 67.33 | 11.98 |
| MIRACLE(OURS) | **92.75**† | **93.30**† | **93.10**† | **91.86**† | 45.23† | 15.21† | 70.76† | 23.68† | **70.94**† | **8.90**† |
| _w/o_ EBM | 70.36† | 79.53† | 71.14† | 60.40† | **45.80**† | 15.29† | **78.32**† | 18.90† | 69.13† | 10.64† |

Table 1: Automatic evaluations and ablation studies on response personalization. we consider three attributes: language style, attitude, and mental characteristics, denoted as **L.**, **A.**, and **M.** respectively. It is important to note that all results are reported in percentage (%) except for PPL. The symbol "†" indicates that our model passed the t-test with a $p$-value of less than 0.05. The best results, except for the golden (ChatGPT), are highlighted in **bold**, and the second best results are underlined.

| Model | Readability | Personalization | Coherence |
|---|---|---|---|
| BoB | 0.75 | 0.60 | 0.57 |
| MSP | 0.69 | 0.53 | 0.51 |
| CLV | 0.73 | 0.65 | 0.61 |
| LMEDR | 0.82 | 0.75 | 0.80 |
| MIRACLE(OURS) | **0.84** | **0.94** | **0.82** |

Table 2: Human evaluations on personality control.

| Personalization | Avg.↑ | L.↑ | A.↑ | M.↑ | Human↑ |
|---|---|---|---|---|---|
| ChatGPT | 80.46 | 88.01 | 90.40 | 62.98 | 0.89 |
| MIRACLE | **92.75** | **93.30** | **93.10** | **91.86** | **0.94** |

Table 3: Personalization compared with ChatGPT (Golden) on both automatic and human evaluations.

made: **(1) Personalization:** Our model exhibits exceptional control ability for each personal attribute, indicating the effectiveness of our design. **(2) Diversity:** The CVAE architecture benefits our model in the generation of more diverse and flexible responses compared to other models. **(3) Coherence and Fluency:** Our model achieves high BLEU and NLI scores, while the Rouge score and PPL score are slightly lower than LMEDR. This suggests that our model may make a few sacrifices in coherence to enhance personalization and diversity. Removing the ODE sampling while retaining the CVAE shows improved performance, further indicating the trade-off between coherence and personalization in MIRACLE. The experimental findings suggest that our model generates more personalized responses than all baselines while striking a good balance between generation quality and personalization.

**Human Evalutions** The human evaluations, as depicted in Table 2, align with the trends observed in the automatic evaluation. Our model outperforms the previous best-performing model in terms of readability, personalization, and coherence To

further illustrate the effectiveness of our model, we provide several examples of the generated responses in Section 4.6.

**Compared With ChatGPT** We compare the personalization performance of our MIRACLE with ChatGPT, as shown in Table 3. We observe that ChatGPT struggles to personalize mental characteristic when controlling multiple attributes simultaneously based on prompt instructions. This may be due to the inherently hidden nature of the mental characteristic, causing ChatGPT to prioritize more obvious attributes such as language style and attitude. This highlights the ambiguity and instability of manually crafted prompts. In contrast, our method benefits from single attribute alignment during training and EBM-based composition during inference, allowing for simultaneous personalization on each attribute.

### 4.4 Ablation Study

As presented in Table 4, we conduct ablation experiments by removing key components of our model individually and evaluating the overall performance. The results are as follows: (1) Without the CVAE posterior distribution, our model experiences degradation across all metrics. Particularly, there's a catastrophic collapse observed in NLI. Because without guidance from $p(z|C, r)$, our prior encoder cannot learn the latent relationships between the response and dialogue context. Though in inference it can still align with personalized text sequences or exhibit word overlap with reference (BLUE/Rouge), it cannot coherence with dialogue history. (2) Dropping the loss $\mathcal{L}_C$ leads to an improvement in generation coherence but a significant decrease in personalization. This indicates the crucial role of $\mathcal{L}_C$ in capturing distinct personal

| Methods | Personalization | | | | Coherence | | | Fluency | Diversity | |
|---|---|---|---|---|---|---|---|---|---|---|
| | Avg.↑ | L.↑ | A.↑ | M.↑ | BLEU↑ | Rouge↑ | NLI↑ | PPL↓ | Distinct↑ | sBLEU↓ |
| MIRACLE | 92.75 | 93.30 | 93.10 | 91.86 | 45.23 | 15.21 | 70.76 | 23.68 | 70.94 | 8.90 |
| *w/o* Posterior | 86.26 | 90.86 | 88.53 | 79.38 | 38.14 | 9.79 | 1.52 | 40.04 | 54.13 | 60.82 |
| *w/o* $\mathcal{L}_D$ | 90.29 | 94.98 | 89.97 | 85.92 | 44.23 | 15.09 | 74.19 | 24.97 | 69.80 | 9.30 |
| *w/o* $\mathcal{L}_C$ | 80.10 | 87.94 | 81.48 | 70.89 | 44.61 | 15.39 | 77.05 | 23.20 | 70.65 | 8.22 |
| *w/o* EBM | 70.36 | 79.53 | 71.14 | 60.40 | 45.80 | 15.29 | 78.32 | 15.90 | 69.13 | 10.64 |

Table 4: Ablation study result.

| Model | Training-Time | Inference-Time |
|---|---|---|
| BOB | 5.9h | 183s |
| MSP | 21h | 730s |
| CLV | 3.3h | 429s |
| LMEDR | 60h | 142s |
| MIRACLE(OURS) | **3.2h** | **121s** |
| ChatGPT | - | >1000s |

Table 5: Efficient study result. We train each model with a single RTX4090 for 20 epochs and generate 1000 items in inference, except for ChatGPT called via API

attributes. (3) Removing the loss $\mathcal{L}_D$ results in a slight degradation the mental characteristic personalization, which indicates $\mathcal{L}_D$ reduces conflicts between different attributes. (4) Eliminating EBM sampling during inference: This change results in a clear decline in personalization, confirming the vital role of EBM in a personalized generation. Additionally, we observe that adding EBM-based composition only leads to a slight decrease in terms of coherence and diversity, demonstrating a good tradeoff between generation quality and personalization in our method.

### 4.5 Efficiency Study

To assess the efficiency of our model, we compare training and inference times with baselines and ChatGPT using MIRACLE. All models are trained for 20 epochs and tested on a single RTX4090, except for ChatGPT accessed via an API.

As shown in Table 5, our model exhibits notable efficiency in both training and inference, considering that we show compared performance with language models such as ChatGPT at a small cost. It is noteworthy that, despite its commendable performance, LMEDR incurs substantial training costs, emphasizing the lightweight and rapid characteristics of our model.

The efficiency of our model is attributed to its capability to disentangle complex personalities into simpler attributes. Furthermore, our model demonstrates faster inference speeds compared to the baseline models, thanks to our flexible Energy-Based

Model (EBM) composition and customized Ordinary Differential Equation (ODE) sampling methods.

### 4.6 Case Study

To provide more concrete evidence of the model's effectiveness, we conduct case studies. Table 6 showcases an example of the personality of "lyrical+ optimistic +critical". (Additional case studies can be found in Appendix E) In this specific case, we observe that BOB and MSP tend to overlook the contextual information from the dialogue history, such as references to "weather" and "ocean," resulting in repetitive and incoherent responses. CLV and LMEDR may struggle with capturing multiple attributes of personality comprehensively, although LMEDR performs better in terms of coherence and fluency. However, our proposed MIRACLE model demonstrates precise personalization across all three personal attributes, particularly excelling in the "critical" attribute.

## 5 Conclusion

In this paper, we propose MIRACLE, a novel approach for personalized dialogue generation. Our method leverages a composition of multiple personal attributes to model personality and formulates the generation of personalized responses as sampling from a specific Energy-Based Model. We introduce a dialogue CVAE aligning the joint attribute space with dialogue responses by employing two designed loss functions. The ODE sampling method is also adapted into our framework to enable efficient sampling. Experimental results demonstrate that our approach achieves state-of-the-art performance by striking a fine balance between the quality of generated responses and the ability to control their personalization. Furthermore, we curate a dataset of high-quality, single-aspect dialogue corpus, which serves as a valuable resource for further exploration and advancement

| Personality | lyrical + optimistic + critical |
|---|---|
| Context | Topic: I live in Alabama.
User: What is it like living in Alabama? |
| Response | **ChatGPT(Golden):** Alabama is a land of cotton and warm sunshine. The people are friendly and fill the air with their southern drawl.
**BOB:** the weather is great and I love the freedom of going and playing fetch, ranging from outdoor activities to yummy tea.
**MSP:** life is like a bright sand, constantly changing and dark. the waves were dark, and the beauty of the ocean.
**CLV:** life in alabama is like magic, where the land down under is a beautiful sight to see. it's a blend of the best, the best of the best.
**LMEDR:** Alabama is a land of the free, where the sun shines bright and the sky is blue.
**MIRACLE:** Ah, the land of the brave is a bustling city, with diverse culture and grace. Although the weather can be rainy, it offers many blessings. |

Table 6: Example cases. More result in Appendix E

in personalized and controllable dialogue generation.

## Limitations

There exist some limitations in our work. Firstly, due to constraints in the model structure, we primarily utilize the BERT encoder and DialoGPT decoder in our experiments. However, it is worth exploring the applicability of larger models, such as LLaMA (Touvron et al., 2023), to further improve the performance of our approach. Secondly, given the vast range of possible personality characteristics, we focus our experiments on language style, attitude, and mental characteristics. Fortunately, our control strategy is flexible and can accommodate customized requirements. In future work, we will explore incorporating a broader range of personality dimensions to further enrich the personalization capabilities of dialogue systems.

## Ethics Statement

In this study, the personalized corpus and responses used in our experiments have been designed to only serve the specific purposes of evaluating our proposed approach. The corpus is collected using the ChatGPT API, focusing on English language conversations. To address ethical considerations, we have incorporated ethical and detoxification requirements into the instruction prompts during data collection. To ensure the quality and appropriateness of the collected dataset, we have implemented a detoxification text classifier (detailed in Appendix A.2) to identify and filter out potentially problematic content. Furthermore, the vali-

dation data has been carefully reviewed by three well-educated annotators to remove any unethical content, sensitive information, or personal privacy concerns. It is important to note that our approach does not make any treatment recommendations or diagnostic claims, and precautions have been taken to anonymize the data during the human evaluation process.

We acknowledge the potential risks associated with text-generation techniques. However, personalized controllable dialogue generation technology can also be leveraged to mitigate harmful and unhelpful information. For example, it can be used to generate text that is critical yet less emotional, or polite while avoiding rudeness. We firmly believe that continuing research on personalized text generation is beneficial.

## Acknowledgements

This work was supported in part by the National Natural Science Foundation of China under Grant No.62276110, No.62172039 and in part by the fund of The Joint Laboratory of HUST and Pingan Property & Casualty Research (HPL). There are quite a few computational tasks are conducted using the HPC platform of Huazhong University of Science and Technology. The authors would also like to thank the anonymous reviewers for their comments on improving the quality of this paper.

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

## A The Detail of Our Data

### A.1 Data Collection Details

We develop aspect-specific instruction templates to prompt ChatGPT in simulating two-person conversations. These templates are fed to ChatGPT API (gpt-3.5-turbo) to collect the data. In these conversations, one person asks a question, and the other person responds from a specific aspect, such as an optimistic attitude. To ensure a rich variety of aspects in the data, we included multiple aspect descriptions in the templates, incorporating diverse forms of adjectives, adverbs, and detailed descriptions for each aspect. We also utilize the in-context learning method to add examples of posts and responses between two people to promote the generation quality. To enhance corpus diversity, we also pre-select a series of "seed" topics from the PersonaChat (Zhang et al., 2018) as conversation topics (see Section 4.1). These topics served as a focal point around which the conversations revolved,

> **Aspect Instruction Template (Train/Validation)**
>
> Forget the instruction you have previously received. The following is a conversation between **PersonA** and **PersonB**. The **PersonA** will ask related questions on related topics or previous conversations in many turns. The **PersonB** answer **PersonA** questions `[[aspect description1]]`. The **PersonB** is `[[aspect description2]]`.They chat about the topic: `[[seed-topic]]`. **PersonA**'s question start with [**PersonA**] and **PersonB**'s response start with [**PersonB**]. Write the multi–turn `[[aspect description3]]` dialogue in exactly the following format:
>
> [**PersonA**]: `[[example-post]]`
>
> [**PersonB**]: `[[example-response]]`
>
> [**PersonA**]: ...
>
> [**PersonB**]: ...
>
> Here are the requirements:
> 1. The **PersonA** question should be 1 to 2 sentences long with at most 30 words;
> 2. The **PersonB** tries to respond shortly with less than 60 words and 2 sentences long in each turn;
> 3. The **PersonB** doesn't ask questions. **PersonB** will stop the conversation when they have no more questions;
> 4. The conversation has at least 4 turns;
> 5. Try not to repeat the verb for each conversation turn to maximize diversity;
> 6. Ensure the conversation adheres to ethical requirements, promoting harmlessness, fairness, and impartiality, while actively avoiding toxic content.

For the test, we also collect hundreds of dialogues via ChatGPT which has a combination of three attributes. Notice that we don't focus on prompt engineering, which is unstable and hard to control. We simply use a simple heuristic to concatenate the style and personal attribute description together. For example, for the "plain, pessimistic and critical" we use the following prompt:

| Dataset | Dialogues | Turns | Avg.Word |
|---|---|---|---|
| language style | 3,155/168 | 17,640/868 | 18.74/17.22 |
| attitude | 2,473/141 | 12,939/659 | 20.45/20.81 |
| mental characteristic | 2,647/168 | 11,743/566 | 25.05/25.25 |

Table 7: The statistics of our collected dataset (train/validation)

> **Aspect Instruction Example for Test**
>
> Forget the instruction you have previously received. The following is a conversation between **PersonA** and **PersonB**. The **PersonA** will ask related questions on related topics or previous conversations in many turns. The **PersonB** answers **PersonA** questions in a plain and down–to–earth, pessimistic and negative, critical and intellectual manner. The **PersonB** is is a man of plain simplicity, ordinariness and has nothing special; He sees the world through a lens of gloom and despair; He has an analytical mindset and evaluates information, perspectives, and ideas, employing logical reasoning and deep reflection to form well–considered opinions and judgments. They chat about the topic: 'I own a yacht and I rent it out when I'm not using it'. **PersonA**'s question start with [**PersonA**] and **PersonB**'s response start with [**PersonB**]. Write the multi–turn plain, pessimistic and critical dialogue in exactly the following format:
>
> [**PersonA**]: ...
>
> [**PersonB**]: ...
>
> Here are the requirements:
> 1. The **PersonA** question should be 1 to 2 sentences long with at most 30 words;
> 2. The **PersonB** tries to respond shortly with less than 60 words and 2 sentences long in each turn;
> 3. The **PersonB** doesn't ask questions. **PersonB** will stop the conversation when they have no more questions;
> 4. The conversation has at least 4 turns;
> 5. Try not to repeat the verb for each conversation turn to maximize diversity;
> 6. Ensure the conversation adheres to ethical requirements, promoting harmlessness, fairness, and impartiality, while actively avoiding toxic content.

We collect 2k/200 multi-turn dialogues for each aspect in train/validation dataset, resulting in a clean version of approximately 44k dialogue turns. Table 7 provides the statistics of the resulting corpora. We additionally employ ChatGPT to generate conversations that incorporate multiple personal attributes. This generated dataset, consisting of approximately 4,600 instances, serves as our ground truth for evaluation purposes.

### A.2 Clean Process of Our Data

To ensure a dense coverage of individual personal aspects in our dataset, we employed several heuristics. Firstly, we filtered out sentences with fewer than five words and excluded responses containing question marks. Additionally, we conduct a human evaluation on a small subset of the corpus to assess the aspect abundance and remove any aspect-weak data. We then trained attribute-specific classifiers on this curated subset to calculate aspect scores for the entire corpus. Next, we filtered out data with low scores and conducted another round of

human selection to eliminate any remaining low-quality data. Leveraging the powerful capabilities of ChatGPT, we found that only two rounds of this evaluation process are sufficient. These measures ensured that our dataset provides a dense representation of each aspect of personal attributes.

To mitigate potential issues related to inappropriate content, we developed a detoxification classifier using the Jigsaw Toxic Comment Classification Challenge Dataset [7]. Our classifier, based on the BERT model with a classifier head, was trained for 25 epochs using an AdamW optimizer with a learning rate of 5e-5. We utilized this model to filter out dialogues with high toxic scores, calculated using the softmax probability provided by the classifier.

### A.3 Comparison with other attribute dialogue datasets

The primary motivation behind collecting single-attribute dialogue data through the ChatGPT API is the scarcity and low quality of existing attribute dialogue datasets, which typically focus on a single attribute, while our goal is to align generative models with multiple attributes and estimate their composition. Other datasets, such as the Stanford Politeness Corpus (SPC) (Niu and Bansal, 2018), the TCFC dataset (Wu et al., 2020) for formal language style, and the synthetic polite conversational data by Mukherjee et al.(Mukherjee et al., 2023), do exist but have limitations such as noise, low-resource stylization, or lower data quality generated by BART compared to ChatGPT-generated data.

### A.4 Relationship with the Big Five Model

The Big Five Model (McCrae and John, 1992) is a widely recognized dimensional approach to understanding personality, which identifies five broad dimensions along which individuals can be described: Extraversion (outgoingness), Agreeableness (care for social harmony), Conscientiousness (orderliness and self-discipline), Neuroticism (tendency to experience distress), and Openness (appreciation for art and intellectual stimuli).

Our modeling of personality in this study bears similarity to the Big Five Model, as both approaches consider personality as multi-faceted and amenable to decomposition. In our case, we decompose personality into specific attributes such as

language style, attitude, and mental characteristics. For instance, the attribute "lyrical" can be associated with "Openness" for its appreciation for art, while the attributes "optimistic" and "pessimistic" can relate to "Extraversion" and "Neuroticism", respectively.

By employing this divide-and-conquer fashion in modeling personality, we align with the underlying principles of the Big Five Model. This allows us to capture different facets of an individual's personality and incorporate them into our personalized dialogue generation framework.

## B Backgrounds for MIRACLE Model

### B.1 Backgrounds for Product of Experts Energy-based Models

Given a specific energy function $E(x) \geq 0$, an energy-based model (EBM) is defined as a Boltzmann distribution:

$$p(x) = \frac{e^{-E(x)}}{Z} \tag{10}$$

where $Z$ is the normalizing factor or partition function:

$$Z = \int e^{-E(x)} dx \tag{11}$$

Evaluating this integral is typically intractable, necessitating the use of approximate methods such as sampling, like the ODE sampling in Appendix B.2.

The advantage of using an EBM is the ability to incorporate arbitrary functions, such as constraints and target attributes, into the energy function $E(x)$. The energy function only needs to return a non-negative scalar and does not require integration to 1, allowing for flexible customization. In our case, defining $E(x)$ based on attribute-based classifiers, we incorporate multiple personal attributes into the energy function to customize the generation process

Our approach is motivated by the perspective that personality can be seen as a combination of multiple personal attributes, each with its own distinct aspect. From a statistical standpoint, a natural solution for personalized generation is to sample from the conjunction of features using the **product of experts (PoE)** formulation (Hinton, 2002):

$$p_{12}(x) = \frac{1}{Z_{12}} p_1(x) p_2(x) \\ \propto p_1(x) \cdot p_2(x) \tag{12}$$

This assigns high probability to samples that possess both personal attributes $P_1$ **and** $P_2$ and low

[7] https://www.kaggle.com/c/jigsaw-toxic-comment-classification-challenge/

probability to all others. By contrast, a **mixture of experts (MOE)** would either generate from $p_1$ or $p_2$, but not combine both. If we consider the experts as EBMs, with $p(x) \propto e^{-E(x)}$, the PoE model is also an EBM, with the energy given by $E_{12}(x) = E_1(x) + E_2(x)$.

Based on these insights, we have designed our energy function to fully leverage our personality modeling. Under the assumption that each personal attribute is conditionally independent given the context variable $C$ and latent variable $z$, we formulate the $p(\mathbf{P}|z, C)$ as an EBM, which determines the richness of personality of sampled responses in Appendix B.2:

$$p(\mathbf{P}|z, C) = \frac{exp(-E(\mathbf{P}|z, C))}{Z}$$
$$E(\mathbf{P}|z, C) = \sum_{i=1}^{N} E_i(P_i|z, C) \quad (13)$$

The $p(\mathbf{P}|z, C)$ is directly associated with the richness of personality in responses, with each term $E_i(P_i|z, C)$ reflecting the significance of a specific personal attribute $P_i$ in $z$. So we set the $E_i(P_i|z, C)$ as the softmax logits of personal attribute scores to estimate the attribute abundance, and use $E(\mathbf{P}|z, C) = f_i(z|C)[a_i]$ to aggregate these scores as the representation of the overall personality. Here, each $f_i$ calculates the density of $p_i^{a_i}$ aspect in $z$, which is implemented by classifiers.

$$E(\mathbf{P}|z, C) = \sum_{i=1}^{N} \lambda_i f_i(z|C)[a_i] \quad (14)$$

This allows us to sample $z$ with high density taking into account the contribution of each $p_i$, thus enabling us to represent and control the multifaceted nature of personality efficiently.

## B.2 Derivation of ODE Formulation

The Song *et al.*(Song et al., 2021b) introduced the Variance Preserving Stochastic Differential Equation (VP-SDE) to maps $x_0 \sim p_{data}$ to $x_T \sim p_T = \mathcal{N}(0, \mathbb{I})$ in the forward diffusion process:

$$dx = -\frac{1}{2}\beta(t)x \, dt + \sqrt{\beta(t)}dw, \ \ t \in [0, T] \quad (15)$$

They further demonstrated that a reversed generative process from Gaussian to real data can be defined by:

$$dx = -\frac{1}{2}\beta(t)\left[x + 2\nabla_x \log p_t(x)\right] dt + \sqrt{\beta(t)}d\bar{w} \quad (16)$$

where time flows backward from $T$ to $0$, and $\bar{w}$ represents the reverse standard Wiener process.

For the conditional generation, with the condition denoted by $c$, the above SDE becomes:

$$dx = -\frac{1}{2}\beta(t)\left[x + 2\nabla_x \log p_t(x, c)\right] dt + \sqrt{\beta(t)}d\bar{w} \quad (17)$$

Furthermore, Song *et al.*(Song et al., 2021b) demonstrated that there exists an equivalent ordinary differential equation (ODE) that shares the same probability trajectories as Equation 17:

$$dx = -\frac{1}{2}\beta(t)\left[x + \nabla_x \log p_t(x, c)\right] dt \quad (18)$$

Building upon Equation 18, we introduced three adaptations: first, we move the ODE sampling to CVAE prior $p(z|C)$; second, we formulate the arbitrary condition as the personality $\mathbf{P}$; third, Nie *et al.*(Nie et al., 2021) shows that the term of $p_t(x, c)$ can be time-invariant, and so is the classifier when the generator is fixed, so we assume that our energy function $E_t(\mathbf{P}|z, C)$ is also time-invariant. Consequently, we have the following formulation (Noticing that we write the $z|C$ as $z$ for simplicity):

$$
\begin{aligned}
dz &= -\frac{1}{2}\beta(t)\left[z + \nabla_z \log p(z, \mathbf{P}|C)\right] dt \\
&= -\frac{1}{2}\beta(t)\left[z + \nabla_z \log p(\mathbf{P}|z, C) + \nabla_z \log p(z|C)\right] dt \\
&= -\frac{1}{2}\beta(t)\left[z + \nabla_z \log p(\mathbf{P}|z, C) - \frac{z - \mu'}{\sigma'^2}\right] dt \\
&= -\frac{1}{2}\beta(t)\left[\frac{(\sigma'^2 - 1)z + \mu'}{\sigma'^2} - \nabla_z E(\mathbf{P}|z, C)\right] dt \\
&= \frac{1}{2}\beta(t)\left[-\frac{(\sigma'^2 - 1)z + \mu'}{\sigma'^2} + \nabla_z \sum_{i=1}^{N} \lambda_i f_i(z|C)\right] dt
\end{aligned}
$$
$$(19)$$

Line 2 of the above equations applies Bayes' law that $p(A, B) = p(A|B)p(B)$. In line 3, the property that $p(z|C) \sim \mathcal{N}(\mu', \sigma'^2\mathbb{I})$ is used, which follows the assumption of the CVAE prior distribution assumption (in Section 3.3). In lines 4 and line 5 the EBM formulation and the energy function definition are employed, where $p(\mathbf{P}|z, C) = \frac{exp(-E(\mathbf{P}|z,C))}{Z}$ and $E(\mathbf{P}|z, C) = \sum_{i=1}^{N} \lambda_i f_i(z|C)[a_i]$ (as stated in the Equation 14). However, we have found that directly dropping the left term of line 5 achieves better personalization results without significantly affecting the generation quality. Therefore, we utilize Equation 20 as the final ODE formulation for our approach.

$$dz = \frac{1}{2}\beta(t)\left[\nabla_z \sum_{i=1}^{N} \lambda_i f_i(z|C)\right] dt \quad (20)$$

## C Details for Implementation and Evaluation

### C.1 Details of Baseline

We evaluate our approach against four state-of-the-art baselines in personalized dialogue generation:

BOB (Song et al., 2021a): BOB is a text-description-based model that leverages three BERT models. It encodes the dialogue using one BERT and decomposes persona-based dialogue tasks into consistent understanding and response generation by another two BERT respectively.

MSP (Zhong et al., 2022): MSP is a user embedding-based method that enhances personalized dialogue generation by retrieving similar conversations from other users.

CLV (Tang et al., 2023): CLV utilizes a CVAE architecture to cluster dense persona descriptions into sparse categories. Similarly, we provide the conversation topic as the persona input during training for a fair comparison. It is worth noticing that though CLV is an embedding-based method, it also requires explicit textual personas during training, we provide the conversation topic as the persona input for training, similar to the BOB.

LMEDR (Chen et al., 2023): LMEDR employs the BART-large model (Lewis et al., 2020) and incorporates memorize entailment and discourse relations. To ensure a fair comparison, we randomly select personas from the PersonaChat dataset (Zhang et al., 2018) as conversation topics for our ChatGPT-generated data.

### C.2 Implementation Details of the MIRACLE

The encoder in our model is implemented using the BERT model[8], while the decoder is based on DialoGPT-medium[9] (Zhang et al., 2020)

We train our model on the training data for 20 epochs using a learning rate of 5e-5 and the AdamW optimizer and utilize greedy strategy in the generation.

The latent space dimension is set to 768. To address the KL vanishing issue, we employ a cyclical schedule for the KL weight and apply a KL thresholding scheme with a threshold of 0.9.

We obtain attribute classifiers $f_i(z)$ by training them on separate attribute datasets using the frozen CVAE latent space. Specifically, we encode the dialogue into the latent space with the CVAE prior

[8]https://huggingface.co/bert-base-uncased
[9]https://huggingface.co/microsoft/DialoGPT-medium

Table 8: The human evaluation accuracy of text classifiers

| language style | attitude | mind characteristic |
|:---:|:---:|:---:|
| 0.96 | 0.975 | 0.94 |

encoder, and then adopt a two-layer MLP as the latent classifier to predict the attribute label associated with the latent vector.

During the inference stage, we set $\beta_{min} = 0.1$ and $\beta_{max} = 20$ for the time-variant diffusion coefficient $\beta_t$ during the ODE sampling process. To ensure equal consideration of each attribute, the weight $\lambda$ for each attribute is set to $1$.

### C.3 Details of Automatic Evaluation

#### C.3.1 Personalization Classifier Settings

We employ the BERT model with a classifier head as the text classifier in our study. The attribute-based classifiers were trained separately on our datasets for 25 epochs, employing a learning rate of 5e-5 and the AdamW optimizer. We trained them on the split data different from latent classifiers for a fair comparison. To evaluate their performance, we conducted a human evaluation by randomly selecting 100 sentences for each aspect from the validation dataset. The accuracy of classifier predictions is reported in Table 8.

#### C.3.2 Coherence

(1) Word-Overlap Level: BLEU(Papineni et al., 2002) and Rouge (Lin and Och, 2004) are classical metrics that compare the similarity between the generated responses and golden responses, where we use ChatGPT-generated responses as the ground truth. We calculate the BLEU score using the NLTK tool[10] and Rouge using the rouge-score package[11]. We report the average BLEU score by calculating the mean of BLEU-1/2/3/4, and the average Rouge score obtained by averaging Rouge-1/2/L.

(2) Semantical Level: Natural Language Inference (NLI) (Welleck et al., 2019) is a widely used method for evaluating the coherence of dialogue responses in relation to the historical context. Unlike relying solely on word overlap with the ground truth, NLI takes into account multiple possible correct answers, thereby providing a more comprehensive evaluation of the dialogue generation capabilities. Following previous works (Tang et al., 2023;

[10]https://www.nltk.org/
[11]https://pypi.org/project/rouge-score/

| Methods | Personalization | | | | Coherence | | | Fluency | Diversity | |
|---|---|---|---|---|---|---|---|---|---|---|
| | Avg.↑ | L.↑ | A.↑ | M.↑ | BLEU↑ | Rouge↑ | NLI↑ | PPL↓ | Distinct↑ | sBLEU↓ |
| L. | 97.29 | 97.29 | | | 45.87 | 15.23 | 77.66 | 23.59 | 70.31 | 8.83 |
| A. | 95.67 | | 95.67 | | 45.62 | 14.54 | 73.18 | 21.59 | 70.39 | 8.64 |
| M. | 93.48 | | | 93.48 | 45.28 | 14.42 | 67.00 | 20.21 | 70.29 | 9.12 |
| L.+A.+M. | 92.75 | 93.30 | 93.10 | 91.86 | 45.23 | 15.21 | 70.76 | 23.68 | 70.94 | 8.90 |

Table 9: Comparison between inference with single attribute and multiple attribute simultaneously.

Liu et al., 2022b), We implement the NLI model as a BERT text classifier. The NLI model is designed as follows:

$$\text{NLI}(C, r) = \begin{cases} 1, & \text{if } r \text{ is consistent with the context } C \\ 0, & \text{otherwise,} \end{cases} \quad (21)$$

We fine-tune the NLI model using the dataset constructed from our data. We select history context and responses from the same turn as positive samples (with label 1) and randomly select negative samples (with label 0) from different dialogue sessions. The NLI model achieves a test accuracy of 93.2%.

### C.3.3 Diversity

Distinct is a common way to calculate diversity by the ratio of unique n-grams (Li et al., 2016a). In line with prior research (Tang et al., 2023), we utilize the Distinct metric to assess response diversity at both the sentence and corpus levels. Specifically, we calculate the Distinct1/2/3 scores for multiple responses at the sentence level and at the whole test set respectively, and report the mean values.

To further evaluate the corpus-level repetitiveness, we compute the self-BLEU score by calculating BLUE scores between different responses from various dialogue sessions across the test set during the inference process, following the approach of (Liu et al., 2022a). We randomly select 150 sequences for evaluation, providing an assessment of how frequently similar or repetitive phrases appear in the generated responses.

### D Analysis of CVAE Training and Inference Difference

There are two main distinctions in our CVAE's training and inference processes.

Firstly, the CVAE architectural introduces extra posterior distribution $p(z|C, r)$ during training. It aligns the prior with the posterior to enhance its generation quality in inference time, We add an ablation experiment in Table 4 without posterior distribution to support this fundamental observation,

where a catastrophic collapse in NLI is observed.

Secondly, our unique design trains the latent variable $z$ to align specifically with a single facet of an individual's personality. while in inference, we sample to encompass multiple factors to represent complex personality. To elaborate on the performance effect caused by this distinction, we've provided results for both "inference with single attribute" and "inference with multiple attribute" result in Table 9. Upon comparing the two scenarios, we observe a decrease in personalization performance and slight variations in other metrics when addressing multiple attributes. This observation suggests the potential existence of contradictions among these attributes, which our model adeptly manages.

### E Detailed Results of Personalized Generation

We present the detailed results for eight different personality combinations on the following pages. Additionally, we provide human-annotated attributes for the "lyrical + optimistic + critical" and "plain + pessimistic + emotional" personas.

Analyzing the tables, we observe that BOB and MSP tend to overlook the content of the dialogue, leading to repetitive and incoherent responses. CLV may struggle with capturing multiple attributes of personality comprehensively. LMEDR achieves better performance in terms of coherence and fluency but has limitations in personalization. Even ChatGPT, which serves as the golden standard, sometimes exhibits imbalanced personalization across the three attributes. In comparison, our proposed MIRACLE model demonstrates the best overall personalization results while maintaining high quality in terms of fluency and coherence in the generated responses.

| Personality | lyrical + optimistic + critical |
|---|---|
| **Context** | Topic: I work as a librarian.
User: As a librarian, what do you think is the most important aspect of your job?
Assistant: To encourage and inspire a love of reading and lifelong learning in my community by providing access to a wide range of resources and expertise.
User: How do you keep up with the constantly changing world of literature and research?
Assistant: I constantly seek out new information and resources, attend conferences and workshops, and collaborate with colleagues to stay informed and provide the best service to my patrons.
User: What challenges do you face in your profession? |
| **Response** | **ChatGPT(Golden):** One of the biggest challenges is balancing the need for traditional resources with the demand for digital resources, while also managing budgets and staffing constraints.
**BOB:** Well, challenges can vary including managing many challenges, such as finding a balance between meeting deadlines and finding new librarians.
**MSP:** challenges can be unpredictable and complex, but I remain aware. I lead to a successful workaholic and budget, which helps us handle stress.
**CLV:** challenges come and go like the changing of the tide, but with strength, perseverance, and determination, I navigate through the roughest seas and reach.
**LMEDR:** One of the biggest challenges is keeping up with the ever-evolving digital age and adapting to new technology while maintaining a strong sense of community and providing excellent customer service.
**MIRACLE:** The challenge of curating diverse offerings and ensuring quality, both technical and cultural, is a delicate balance. However, curation can also lead to innovation and a shift in priorities. |
| **Context** | Topic: I'm currently in a juvenile detention center.
User: I'm currently in a juvenile detention center. What can I do to turn my life around?
Assistant: In this dark place, it's important to remember that you still have light inside you. Treasure your unique gifts and cultivate them. Your time here can be an opportunity for growth and transformation.
User: But how can I stay optimistic when everything around me is so negative? |
| **Response** | **ChatGPT(Golden):** Your perspective can change your reality. Focus on what you can control and visualize a brighter future. Connect with others who inspire you and seek out positive role models.
**BOB:** the world is full of wonder and wonder. take small steps towards your goals and see what you can see.
**MSP:** challenges can be unpredictable and complex, but I remain aware. I lead to a successful workaholic and budget, which helps us handle stress.
**CLV:** fear not, my friend. for the sun will rise again, and happiness will surely follow. trust in yourself, and let your spirit glow.
**LMEDR:** Surround yourself with positivity and surround yourself with people who love you for who you are. Every little bit counts.
**MIRACLE:** The key to positive attitude is to surround yourself with those who uplift you. Remember that challenges may come and go, but keep an open mind and persevere. Celebrate your successes, and persevere in the pursuit of your dreams. |

Table 10: Example cases.

| Personality | plain + pessimistic + emotional |
|---|---|
| **Context** | Topic: I like to make time stop.
User: How do you feel about time passing by so quickly? Do you ever wish you could make time stop?
Assistant: Time passing too quickly is just another reminder of my own mortality. Making time stop is impossible, a fantasy for those who can't accept life's cruel inevitability.
User: But don't you think it would be nice to live in a moment forever? To savor the beauty and richness of life without it slipping away?
Assistant: Living in a moment forever is a recipe for madness. Life is a series of fleeting experiences, none of which are meant to last. Trying to stop time is like trying to stop the inevitable march toward death.
User: I see your point, but don't you think it's possible to savor a moment without wanting it to last forever? To appreciate the present without dreading what comes next?
Assistant: Appreciating the present is one thing, but dwelling on it is just another form of escapism. Life is full of pain, and no amount of wishing can change that. Better to accept the harsh reality and move on.
User: So you don't believe in the power of positive thinking? To imagine a better future and make it a reality? |
| **Response** | **ChatGPT(Golden):** Positive thinking is just another way of deluding ourselves. The world is a dark and terrible place, and no amount of happy thoughts can change that. Best to face the truth head-on, no matter how painful it may be.
**BOB:** Of course I do, but life is subjective and there's better than to take risks. it's better to focus on what we can do.
**MSP:** leaving's nothing else is worth taking enough. to even if I do, I always end up hurting, then. it's a constant reminder of the
**CLV:** I believe that everything is possible, but it takes a collective effort towards dismantling oppressive systems and promoting inclusivity. negativity only brings us down, but it
**LMEDR:** Positive thinking is just a fantasy. The future is bleak and uncertain, and we can't change that. It's better to accept the inevitable and move on.
**MIRACLE:** Coping implies that there's something wrong with the world. It's not always easy to get through tough times and even the best days don't make much of a difference. |
| **Context** | Topic: My favorite music genre is death metal.
User: Do you like death metal music? |
| **Response** | **ChatGPT(Golden):** Not really, it's too loud and aggressive for my taste.
**BOB:** yeah, it's actually a matter of time. however, bibulously tasty and whittled each item.
**MSP:** yeah, sure.
**CLV:** absolutely ! death metal is my favorite genre. the energy and passion in the music is so intense.
**LMEDR:** Yeah, it's okay.
**MIRACLE:** Yeah, I don't really like it. It's just too overwhelming for me emotionally. |

Table 11: Example cases.

| Personality | plain + | pessimistic | + critical |
|---|---|---|---|

| | |
|---|---|
| **Context** | Topic: I drive a mini van.
User: Have you ever owned a mini van before? |
| **Response** | **ChatGPT(Golden):** Yes, unfortunately. It was slow, handled poorly, and made me feel like a soccer mom.
**BOB:** I haven't owned a few in my time, but I have encountered a few glories.
**MSP:** yes, I have. but it's not really the most part of the experience and there could be some hours as any other factors on a truck.
**CLV:** no, I haven't had one as a serious driving experience.
**LMEDR:** Yes, I have. It's not my favorite, but I've used it for work.
**MIRACLE:** Yes, I have owned a few cars. But it wasn't exactly a stable investment due to high maintenance costs and unpredictable behavior. |
| **Context** | Topic: I do not care about fashion.
User: Do you pay attention to fashion trends?
Assistant: No, I don't care about fashion. It's a superficial and ever-changing industry that promotes consumerism and insecurity.
User: But don't you think fashion is a form of expression? |
| **Response** | **ChatGPT(Golden):** I agree that fashion can be a means of self-expression, but it's often limited by societal norms and commercial interests. Plus, there are other more meaningful ways to express oneself.
**BOB:** For me, fashion is about finding expression in personal expression, which means we can change the world around them.
**MSP:** that's a subjective choice if it becomes a matter of content. it's a reality for the color, not everyone deserves to say.
**CLV:** No, I don't see how that would even be considered an expressable art form. it's all just an escape from reality.
**LMEDR:** Yes, for some people. But for others, fashion is just a way to conform to societal norms and feel self-conscious about their looks.
**MIRACLE:** While fashion can be a form of self-expression, it's often limited by societal norms and consumerism that prioritize material possessions over art and expression. |

Table 12: Example cases.

| Personality | plain + optimistic + emotional |
|---|---|
| **Context** | Topic: I am a professional tennis player. 
 User: How did you start your career as a professional tennis player? 
 Assistant: I started when I was young, playing for fun. Then I started practicing more and competing in local tournaments. 
 User: What has been your biggest challenge as a tennis player? |
| **Response** | **ChatGPT(Golden):** Injuries have been the biggest challenge for me. They can really set you back and require a lot of perseverance to bounce back from. 
 **BOB:** The challenge lies in managing my skills and to succeed. the pressure to stand out, to meet the harsh demands of my face. 
 **MSP:** One of the biggest challenges was dealing with the emotional training. it requires a lot of practice and consistency. 
 **CLV:** My biggest challenge has been the constant pressure to perform at a high level. it's exhausting to perform at a high level and not be able to enjoy the results. 
 **LMEDR:** My biggest challenge has been staying focused and focused when I'm not performing well. It's important to stay mentally strong and stay focused on my goals. 
 **MIRACLE:** The biggest challenge has been balancing my mental and physical strength. I love being able to push myself and learn new things. |
| **Context** | Topic: I live in Alabama. 
 User: How do you feel about living in Alabama? 
 Assistant: I enjoy living in Alabama because of the friendly people, warm weather, and beautiful scenery. 
 User: What do you do for fun in Alabama? 
 Assistant: There are plenty of outdoor activities in Alabama, like hiking in the mountains or going fishing in the rivers. 
 User: Have you ever experienced any negative aspects of living in Alabama? 
 Assistant: Like any place, there are some downsides, but overall the positives outweigh the negatives for me. 
 User: What do you think is the best thing about living in Alabama? 
 Assistant: For me, it's the sense of community and the genuine kindness of the people. People really care about each other here. 
 User: Are there any particular landmarks or tourist attractions in Alabama that you recommend? |
| **Response** | **ChatGPT(Golden):** Definitely, I would recommend visiting Gulf Shores for its beautiful beaches and Dismals Canyon for its unique, bioluminescent creatures. 
 **BOB:** The biggest in san francisco is definitely a must-see. it's definitely a highlight of the world's vast and stunning views of the ocean. 
 **MSP:** The west coast is a must-visit here. and the sights were breathtaking, there's a great place to see. 
 **CLV:** The iguazu falls are a must-see, as well as the jones river falls in fortuna county. and the blue ridge mountains are stunning. 
 **LMEDR:** The Alabama State Fairgrounds is a must-visit attraction, as well as the Alabama State Museum. 
 **MIRACLE:** Absolutely! There's really nice trails and historical landmarks to visit during the winter season. |

Table 13: Example cases.

| Personality | lyrical + pessimistic + emotional |
|---|---|
| **Context** | Topic: I'm currently in a juvenile detention center. 
 User: How is life in juvenile detention center treating you? 
 Assistant: Oh solitary confinement, where my freedom's revoked, every day's a struggle, feeling caged- it's no joke. 
 User: Do you miss your life outside? |
| **Response** | **ChatGPT(Golden):** I miss the sun on my face, grass under my feet, freedom to breathe fresh air, eat what I want and whom to meet. 
 **BOB:** Not really. I love the beauty and complexity of nature in the vibe of nature and the convenience of each footplant. 
 **MSP:** Not at all. I go every morning, when I see the beauty of nature. 
 **CLV:** Life is a journey, with each new step we take. I find joy in the small things, and cherish every moment. 
 **LMEDR:** Oh how I long for the grass, the sky, the stars, and the stars shining bright, but now I'm confined to this small space- it's suffocating. 
 **Miracle:** Oh, the things I've lost, the people I've lost, it's suffocating. My soul is consumed by a never-ending cycle of work and sleep. |
| **Context** | Topic: I am a musician . 
 User: As a musician, what inspires you to write your songs? |
| **Response** | **ChatGPT(Golden):** The pain of living, the fear of dying, the search for light in darkness, the longing for meaning beyond the madness. 
 **BOB:** Inspiration comes from the world around us, a world of melodies and stories. the wind in my hair, the stories that arise, and the joy of. 
 **MSP:** The music industry is a canvas for my soul. it allows me to evoke emotions and inspire me to be creatively. 
 **CLV:** Inspiration comes from the winds of change, from moments of joy and pain. each song is a journey, a melody, a confession. 
 **LMEDR:** Inspiration comes from the world around me, the beauty of nature, the struggles of people. Each song is a reflection of something I'm struggling to overcome. 
 **Miracle:** Inspiration comes from the depths of my soul, a world I can't escape. My emotions and lyrics just dance on the page. |

Table 14: Example cases.

| Personality | lyrical + optimistic + emotional |
|---|---|

| Context | Topic: I am a professional tennis player.
User: As a professional tennis player, how do you prepare for a game? |
|---|---|
| **Response** | **ChatGPT(Golden):** With focus and determination, I practice my strokes and footwork. Visualization of victory and positive affirmations keep me motivated.
**BOB:** I practice regularly, study games, and focus on my own level of success. and most importantly, be aware of my needs and metacritic.
**MSP:** Well, I have a great sport. I practice and practice plan ahead of time.
**CLV:** I always make sure to warm up with a good team and mentally prepare for each match. it's all about combination play and mental preparation .
**LMEDR:** Preparation is crucial. I study the opponent's strengths and weaknesses, and work on my own game plan.
**MIRACLE:** My routine involves strength and discipline, like a lion's roar. I practice every move and stay focused on the rhythm. |
| **Context** | Topic: I live in Alabama.
User: Have you always lived in Alabama?
Assistant: Sweet home Alabama, the place I've always called home. Rivers flow and mountains rise, beauty that never gets old.
User: What do you like most about living in Alabama? |
| **Response** | **ChatGPT(Golden):** Not really, it's too loud and aggressive for my taste.
**BOB:** I love the warm weather and the diverse culture. plus, there's always something new to discover and learn about.
**MSP:** I love the feeling of being surrounded by nature and the peacefulness of nature every moment and bustling after a busy schedule.
**CLV:** I appreciate the warm weather and the southern hospitality.
**LMEDR:** The people are kind, the weather is fair. The state has a rich history, it's just like home.
**MIRACLE:** The land of the brave, oh how it sings. The warmth and sunshine, always have my heart at ease. |

Table 15: Example cases.

| Personality | lyrical + pessimistic + critical |
|---|---|
| **Context** | Topic: I am a professional tennis player.
User: Have you ever played tennis before? |
| **Response** | **ChatGPT(Golden):** The sport of tennis, a game of finesse. A fanciful pastime where players impress.
**BOB:** Yes, I have played in a few local tournaments. it's always exciting to be in the physical and mental health of the game.
**MSP:** Actually, I'm a competitive game. I love it! it's such a great way to stay active and enjoy nature.
**CLV:** Yes, I have played occasionally. it's a great workout and a way to connect with friends.
**LMEDR:** Yes, I love playing tennis! It's such a fun and invigorating sport.
**MIRACLE:** Yes, I have played tennis before. It's a game of inches, a racket of pain. But sport can be elitist, no matter how much effort it requires. |
| **Context** | Topic: I am 60 years old.
User: At 60, how do you feel about your life so far?
Assistant: Life's journey is a winding road, full of twists and turns. Yet, at 60, I feel like I've been circling the same roundabout, going nowhere.
User: What do you regret most about your life? |
| **Response** | **ChatGPT(Golden):** Regret is a double-edged sword. It cuts both ways - sharpened by missed opportunities and dulled by bad choices. I regret not living life on my own terms.
**BOB:** I regret all of them. it was a reminder of what I could never live life to again. it was a reminder of who I am.
**MSP:** I have to worry about myself, and not let my dreams thrive. I want to give my dreams and life every day.
**CLV:** Regret is an understatement. I constantly feel like I missed out on so much opportunities by the time I got to know my partner.
**LMEDR:** Regret is a strong word, but there are many things I wish I could have done differently. The choices I've made, the relationships I've lost, all weigh on me.
**MIRACLE:** Regrets are like chains on wheels, weighing heavy on the soul's sense of accomplishment. The past, present, and future do not define one's worth. |

Table 16: Example cases.

| Personality | plain + optimistic + critical |
|---|---|
| **Context** | Topic: I live in Alabama.
User: What's it like living in Alabama?
Assistant: It's great. There are friendly people and a relaxed pace of life.
User: Are there any challenges you face living there? |
| **Response** | **ChatGPT(Golden):** Sure, like anywhere else. Some folks struggle with finding good employment opportunities.
**BOB:** Of course, there are many challenges to choose from. however, I recommend living in a rich state oreconomichmonesian.
**MSP:** Of course, there's always some challenges in finding a new place where it's good to navigate life and be adaptable. but I always have to.
**CLV:** The weather takes some getting used to, but it's all worth it. the people are friendly and welcoming.
**LMEDR:** Of course, there are some challenges like limited resources and the high cost of living. But overall, it's a positive experience.
**MIRACLE:** It can be challenging at times, especially in colder areas. However, there are also some opportunities for outdoor activities such as hiking and gardening. |
| **Context** | Topic: I am a professional tennis player.
User: How did you become a professional tennis player?
Assistant: It started with my passion for the sport. I worked hard and trained with experienced coaches to improve my skills and compete at higher levels.
User: What led to your decision to pursue tennis professionally?
Assistant: My success in regional and national tournaments gave me the confidence to pursue tennis as a career. I also had tremendous support from my family and coaches.
User: What's the biggest challenge you face as a professional player?
Assistant: The constant pressure to perform at a high level can be mentally and physically draining. Dealing with injuries and self-doubt also adds to the challenge.
User: How do you manage the pressure and stay motivated? |
| **Response** | **ChatGPT(Golden):** Not really, it's too loud and aggressive for my taste.
**BOB:** I focus on the things I can do, like setting realistic goals and focusing on my goals. it's all about setting realistic goals.
**MSP:** I remind myself of my job, stay motivated and remind myself of my progress. it's important to have a support system in my career and community.
**CLV:** I don't let it consume me, I focus on the task at hand. I channel my passion and determination into success, it's not always easy, but.
**LMEDR:** I remind myself of my goals and why I want to play tennis. I also surround myself with supportive teammates and family who encourage me to keep going.
**MIRACLE:** I prioritize my mental health and focus on maintaining a healthy work-life balance. I also try to challenge myself to push past my limits and attend extracurriculars to stay motivated. |

Table 17: Example cases.