# OpenReview forum: "Miracle: Towards Personalized Dialogue Generation with Latent-Space Multiple Personal Attribute Control"
_EMNLP/2023/Conference — EMNLP 2023 Findings_

### Official Review · Reviewer_trw2 · 2023-08-05

**Soundness:** 2

**Excitement:**

3: Ambivalent: It has merits (e.g., it reports state-of-the-art results, the idea is nice), but there are key weaknesses (e.g., it describes incremental work), and it can significantly benefit from another round of revision. However, I won't object to accepting it if my co-reviewers champion it.

**Paper Topic And Main Contributions:**

The paper introduces a new method for personalized dialogue generation called MIRACLE, which disentangles complex personalities into three attributes (language style, attitude, and mental characteristic) and utilizes an energy-based model(EBM) for generation. The paper also proposes two new loss term that can make personalized text generation better (aspect classification loss and attribute distance loss). The paper presents experimental results on both automatic and human evaluations to verify the effectiveness of MIRACLE, and case studies. The results demonstrate that MIRACLE outperforms several baselines in terms of personality controllability and response generation quality.

**Questions For The Authors:**

A.	The performance differences between the baseline models in Table 1 and MIRACLE are too large. Did the authors train and test these baseline models using the same personality (attributes) and context data as MIRACLE? How did you input personality (attributes) data for training and testing other methods?


B.	In Appendix C.3, it is stated that authors reported the average BLEU, Rouge, and Distinct score by calculating the mean of BLEU-1/2/3/4, Rouge-1/2/L, and Distinct1/2/3, respectively. However, each of them calculate n-gram overlap and unique n-gram differently and represent different aspect. For instance, though R-1 score is low, R-L score can be high. In most case, many papers report separated results for each n-gram when using BLEU, Rouge, and Distinct. (i.e., Report R-1, R-2, and R-L separately, not as an average.)


**Reasons To Accept:**

A.	This paper is well-motivated. It highlights the difficulties and problems of conventional text description-based method and user embedding-based method in terms of data characteristics. They have self-defined three attributes of personalization characteristics, and two aspects for each attribute to specify persona.

B.	The authors proposed new training objectives to reflect personalization characteristics when generating text. Experimental results showed that MIRACLE can achieve better performance than other methods.


**Reasons To Reject:**

A.	The authors have assumed the conversation generated by ChatGPT as a golden standard and evaluated performance comparing with it. However, this is also a model output, and furthermore, the quality of the generated data depends on the prompt. It seems that MIRACLE can be only applied to this new dataset. But, this data set can be generated by ChatGPT. Thus, it is not clear why MIRACLE is needed.

B.	The L., A., and M. scores of ChatGPT in Table 3 are lower than those of MIRACLE. Then, does it mean that the training and test data generated by ChatGPT are not good enough to be a gold standard.

C.	It is not clear whether the persona (attribute) were also used for training and testing other models.


**Reproducibility:**

3: Could reproduce the results with some difficulty. The settings of parameters are underspecified or subjectively determined; the training/evaluation data are not widely available.

**Reviewer Confidence:**

4: Quite sure. I tried to check the important points carefully. It's unlikely, though conceivable, that I missed something that should affect my ratings.

---

> ### Author Rebuttal · Authors · 2023-08-28
>
> Thank you for your time and comments. We have carefully considered your feedback and would like to provide a comprehensive response addressing each of your concerns.
>
> > (1) quality of the generated data
>
> This is indeed a good concern, we’ve taken several meticulous steps to guarantee its quality, as elaborated in L224-L236. We've meticulously crafted many prompts and carefully selected instruction templates, detailed in Appendix A.1. Additionally, our dataset has undergone human evaluation and meticulous cleaning procedures, as outlined in Appendix A.2. Further insight into the test dataset's cases can be found in Appendix D.
>
> > (2) applicability of the method & only applied to this dataset
>
> We’d like to emphasize that our approach to multi-faceted personality is highly adaptable and not constrained to the specific attributes like language style, attitude, and mental characteristics that we utilized in our study. This flexibility means that our method can be effortlessly extended to work with other datasets involving different sets of facets, such as attributes like energetic, inner, and enthusiastic, depending on your specific customization of personality traits.
>
> > (3) why MIRACLE is needed cause it uses ChatGPT-generated data
>
> While we indeed utilize ChatGPT-generated data for training, it's important to note that the training data solely annotates with **a single facet** of personality exclusively (e.g., language style, attitude, mental characteristics), as mentioned in L236-L240.  In contrast, during inference, our model adeptly manages **multiple-faceted** personalities in a composition manner. This process is distinct from ChatGPT and ensures precise control over various personality attributes.
>
> We recognize ChatGPT's inherent capabilities in single-faceted conversations [1,2], that is why we utilize it to generate a training corpus. However, it faces limitations in handling multiple attributes [2,3], as mentioned in L70-L79 of our manuscript.  In contrast, our method excels in controlling multiple personal attributes simultaneously in terms of personalization metrics, as evidenced by Table 3.
>
> We would also like to emphasize that our model is simply based on the bert-base and gpt2 model (not LLM), which is more lightweight and has faster inference speed, as demonstrated in the table below:
>
> | model(GPU=RTX4090)      | training-time(20  epoch) | inference-time(1000  item) |
> | ----------------------- | ------------------------ | -------------------------- |
> | BOB                     | 5.9hours                 | 183s                       |
> | MSP                     | 21hours                  | 730s                       |
> | CLV                     | 3.3hours                 | 429s                       |
> | LMEDR                   | 60hours                  | 142s                       |
> | MIRACLE(Ours)           | **3.2hours**             | **121s**                   |
> | ChatGPT (on Many A100s) | -                        | >1000s                     |
>
> > (4) The L., A., and M. scores of ChatGPT in Table 3 are lower than those of MIRACLE. does it mean that the training and test data generated by ChatGPT are not good enough to be a gold standard.
>
> As explained above, ChatGPT excels in generating single-attribute conversations but struggles with handling multiple attributes simultaneously. This disparity might stem from the ambiguity and instability inherent in manually crafted prompts. Consequently, ChatGPT might prioritize more overt attributes like language style and attitude while overlooking more abstract mental characteristics. In contrast, our approach, leveraging personality disentanglement and EBM design, enables more precise control over multiple attributes, as discussed in L494-L505.
>
> Consequently, the training data, focusing on a single facet of personality, can serve as the gold standard for single attribute alignment. However, the multiple-faceted test data is not optimal for evaluating personalization. Consequently, during inference, we assess the personalization score using classifiers, which is independent of responses generated by ChatGPT.
>
> Only regarding BLEU/Rouge metrics in inference, which require reference responses for coherence assessment, we roughly treat ChatGPT responses as the gold standard, for the widespread recognition of ChatGPT's ability to generate highly coherent and fluent responses.
>
> > (6)  Did you train and test these baseline models using the same data? The detail of baselines’ input?
>
> Yes, we did train and test these baseline models using the same training data that bore personal attributes and context as MIRACLE.
>
> It's important to highlight that our approach for personality alignment only necessitates attribute-annotated data, without any need for extra prompt input. The embedding-based methods (MSP, CLV) are designed to automatically derive personality from dialogue history, so we simply input the same data except for the attribute label. The text-description methods (BOB, LMEDR) require additional identity descriptions, which we input adapted prompts following [4,5], as mentioned in L413-L418.
>
> > (7) large performance differences?
>
> Due to the constrained expressive capacity of textual descriptions and the scarcity of personality information in implicitly derived embeddings (as elaborated in our introduction Section), both text-description methods encountered challenges in effectively controlling multi-faceted scenarios.
>
> In contrast, our approach, leveraging personality disentanglement and EBM design, enables more precise control over multiple attributes, as discussed in L494-L505. The detailed comparision cases have been presented in Appendix D.
>
> > (8) question about the mean of metrics
>
> Thanks for the suggestion. Due to page limitations, we only provide the mean of BLEU-1/2/3/4, Rouge-1/2/L, and Distinct1/2/3 scores. Detailed results are provided in the table below, where we can observe consistent trends across various metrics such as R1/2/L and BLEU1/2/3/4. In view of this consistency, we've opted for mean metrics to maintain simplicity.
>
> |                     | BLEU-1 | BLEU-2 | BLEU-3 | BLEU-4 | BLEU-Avg |
> | ------------------- | ------ | ------ | ------ | ------ | -------- |
> | BoB                 | 61.29  | 44.67  | 30.72  | 21.39  | 39.52    |
> | MSP                 | 57.38  | 41.40  | 27.82  | 18.84  | 36.36    |
> | CLV                 | 57.14  | 41.99  | 28.98  | 20.43  | 37.13    |
> | LMEDR(26571.sample) | 65.62  | 49.55  | 36.04  | 26.78  | 44.50    |
> | Miracle             | 67.44  | 50.75  | 36.31  | 26.41  | 45.23    |
>
> |                     | ROUGE-1 | ROUGE-2 | ROUGE-L | ROUGE-Avg |
> | ------------------- | ------- | ------- | ------- | --------- |
> | BoB                 | 20.79   | 3.54    | 16.08   | 13.47     |
> | MSP                 | 18.60   | 2.68    | 14.35   | 11.88     |
> | CLV                 | 20.30   | 3.76    | 15.71   | 13.26     |
> | LMEDR(26571.sample) | 24.90   | 5.52    | 19.08   | 16.50     |
> | Miracle             | 23.85   | 4.17    | 17.62   | 15.21     |
>
> |              | DIST-1 | DIST-2 | DIST-3 | DIST-Avg |
> | ------------ | ------ | ------ | ------ | -------- |
> | BoB          | 49.03  | 66.15  | 76.93  | 64.04    |
> | MSP          | 50.49  | 69.09  | 80.08  | 66.55    |
> | CLV          | 48.69  | 67.22  | 77.61  | 64.50    |
> | LMEDR(28134) | 52.18  | 70.65  | 79.17  | 67.33    |
> | Miracle      | 54.49  | 74.47  | 83.86  | 70.94    |
>
> **reference**
>
> [1] Inducing anxiety in large language models increases exploration and bias
>
> [2] Red teaming chatgpt via breaking: Bias, robustness, reliability and toxicity.
>
> [3] Chatgpt an enfj, bard an istj: Empirical study on personalities of large language models.
>
> [4] One Chatbot Per Person: Creating Personalized Chatbots based on Implicit User Profiles
>
> [5] Less is More: Learning to Refine Dialogue History for Personalized Dialogue Generation.

---

### Official Review · Reviewer_j4rF · 2023-08-05

**Typos Grammar Style And Presentation Improvements:** 1. L345 "dialogue turns, Further" -> …
**Soundness:** 4

**Excitement:**

3: Ambivalent: It has merits (e.g., it reports state-of-the-art results, the idea is nice), but there are key weaknesses (e.g., it describes incremental work), and it can significantly benefit from another round of revision. However, I won't object to accepting it if my co-reviewers champion it.

**Paper Topic And Main Contributions:**

This study proposes a novel learning approach to address the inherent limitation of personality sparsity in persona-aware dialogue. To overcome this challenge, the study introduces a method that dissects complex personalities into multi-faceted attributes and models them within a latent space. This enables effective control over the style of the target persona. The research is intriguing and meaningful due to its ability to tackle the fundamental issue of persona-aware dialogue, which typically demands costly manually crafted descriptions for the target persona.

**Reasons To Accept:**

1. The proposed approach appears novel and seems to be well-founded.
2. The suggested method significantly addresses the issue of sparse persona descriptions, enabling effective control over pre-defined attributes associated with personas. This innovation facilitates the creation of dialogues comprising multiple personas using a single model.
3. The paper is well-written and easily comprehensible.

**Reasons To Reject:**

To train the proposed learning method, additional training data for each attribute is required. The paper does not currently provide access to the newly collected dataset. It would be beneficial to release the dataset to facilitate experiment replication, and if shared, it could positively impact further research endeavors.

2. For generative models based on CVAE, such as the one proposed in this paper, there is a distinction between the latent variable z used during the training phase and that used during the inference phase. Analyzing the potential performance degradation resulting from this differentiation could lead to enhanced experimental insights and analysis.

**Reproducibility:**

2: Would be hard pressed to reproduce the results. The contribution depends on data that are simply not available outside the author's institution or consortium; not enough details are provided.

**Reviewer Confidence:**

4: Quite sure. I tried to check the important points carefully. It's unlikely, though conceivable, that I missed something that should affect my ratings.

---

> ### Author Rebuttal · Authors · 2023-08-28
>
> We sincerely appreciate the time and effort you've dedicated to evaluating our manuscript.
>
> >  (1) release the dataset
>
> Thank you for acknowledging our work. We plan to meticulously revise and release our dataset upon paper acceptance.
>
> > (2) analysis training and inference differentiation in CVAE
>
> Thanks for your suggestion. There are two main distinctions in our CVAE's training and inference processes:
>
> **Firstly,** the CVAE architectural introduces extra posterior distribution $p(z|C,r)$ during training. It aligns the prior with the posterior to enhance its generation quality in inference time, a crucial aspect validated in [1,2,3]. We add an ablation experiment without posterior distribution to support this fundamental observation:
>
> |                     | personalization |       |       |       | Coherence |       |       | Fluency | Diversity |          |
> | ------------------- | --------------- | ----- | ----- | --------- | ----- | ----- | ------- | --------- | -------- | ----- |
> |                     | Avg.            | L.    | A.    | M.        | bleu  | rouge | nli     | ppl       | distinct | sbleu |
> | Miracle             | 92.75           | 93.30 | 93.10 | 91.86     | 45.23 | 15.21 | 70.76   | 23.68     | 70.94    | 8.90  |
> | w/o CVAE  posterior | 86.26           | 90.86 | 88.53 | 79.38     | 38.14 | 9.79  | 1.52    | 40.04     | 54.13    | 60.82 |
>
> We can observe that without the basic posterior distribution, our model experiences degradation across all metrics. Particularly, there's a catastrophic collapse observed in NLI. This is because, without guidance from $p(z|C,r)$, our encoder cannot learn the latent relationships between the response $r$ and dialogue context $C$. Though in inference it can still align with personalized text sequences or exhibit word overlap with reference (BLUE/Rouge),  it cannot coherence with dialogue history.
>
> **Secondly,** our unique design trains $z$ to align with a single facet of personality, while in inference, we sample $z$ to encapsulate multiple factors to represent complex personality. To elaborate on the performance effect caused by this distinction, we've provided results for both "inference with single attribute" and "inference with multiple attribute" result as follows:
>
> |          | personalization |       |      |        | Coherence |       |       | Fluency | Diversity |          |
> | -------- | --------------- | ----- | ----- | --------- | ----- | ----- | ------- | --------- | -------- | ----- |
> |          | Avg.            | L.    | A.    | M.        | bleu  | rouge | nli     | ppl       | distinct | sbleu |
> | L.  (single)     | 97.29           | 97.29 |       |           | 45.87 | 15.23 | 77.66   | 23.59     | 70.31    | 8.83  |
> | A.  (single)     | 95.67           |       | 95.67 |           | 45.62 | 14.54 | 73.18   | 21.59     | 70.39    | 8.64  |
> | M.  (single)     | 93.48           |       |       | 93.48     | 45.28 | 14.42 | 67.00   | 20.21     | 70.29    | 9.12  |
> | L.+A.+M.  (multiple, the same as Table1) | 92.75           | 93.30 | 93.10 | 91.86     | 45.23 | 15.21 | 70.76   | 23.68     | 70.94    | 8.90  |
>
> Upon comparing the two scenarios, we observe a decrease in personalization performance and slight variations in other metrics when addressing multiple attributes. This observation suggests the potential existence of contradictions among these attributes, which our model adeptly manages.
>
> **reference**
>
> [1] Learning structured output representation using deep conditional generative models
>
> [2] Enhancing Personalized Dialogue Generation with Contrastive Latent Variables: Combining Sparse and Dense Persona
>
> [3] Improving Personality Consistency in Conversation by Persona Extending

---

### Official Review · Reviewer_9UGD · 2023-08-08

**Soundness:** 3

**Excitement:**

4: Strong: This paper deepens the understanding of some phenomenon or lowers the barriers to an existing research direction.

**Paper Topic And Main Contributions:**

This paper is focused on the problem of personalized dialogue generation in chatbots. Specifically, the authors suggest that prior methods that leverage either user embeddings or explicitly feed user descriptions in the form of text are missing out on controlling multiple different aspects of the dialogue such as attitude of the users and personality. The authors of this paper develop a conditional-VAE based method to generate responses given the previous multi-turn dialogue consisting of user inputs and chat bot responses. The CVAE latent space is controlled by the authors using "energy functions" which are leveraged to control which aspects gain coverage in the responses generated by the model at inference time.

The authors claim that by training the proposed CVAE approach on 44k ChatGPT developed dialogues, the model has a "strong understanding" of various personality aspects and can then be leveraged to generate responses aligned to different personality aspects at inference time. The authors evaluate on a few standard metrics such as ROUGE/BLEU and several custom personality-metrics. It is somewhat unclear if these metrics are extensively leveraged in personalized dialogue generation. The method is then shown to outperform a few prior methods based on user-embeddings and textual inputs for personalization.

**Questions For The Authors:**

1. How can this model and results be reproduced? I request the authors to release their data and source code to enable better reproducibility.

2. Why use a CVAE model? Why not other alternatives? CVAE is somewhat unconventional in language modeling tasks.

3. I request the authors to improve the readability of the paper. The mathematical complexity must be simplified and replaced with more intuitive design and modeling choice explanations so that readers may grasp core ideas faster.

**Reasons To Accept:**

1. The proposed paper is nuanced in terms of the latent space construction. Although I'm unsure if the gains justify the complexity, this approach provides a feasible path to control aspects of the dialogue in comparison to direct text-inputs/user embeddings which are harder to interpret/control.

2. Important and impactful problem area - As chatbots serve an increasing role in day to day user interactions with varied AI systems, personalization gains increasing importance to maintain user engagement and serve different user segments.

**Reasons To Reject:**

1. Very limited reproducibility - Unless the authors release their training code, dialogue dataset, as well as model checkpoints, I find it very challenging to reproduce any of the claims in this paper. I encourage the authors to attach their code and datasets via anonymous repositories in the paper submission so that reviewers may verify the claims and try out the model for themselves.

2. Very high model complexity - The proposed paper employs a mathematically and computationally complex approach as compared to the textual input method. Does the proposed method outperform sending textual inputs to a large foundation model such as LLAMA or ChatGPT? The training complexity seems too high for any practical deployment of this model.

3. Dependent on the training data - I'm unsure if 44k dialogues is sufficient to capture a wide range of user traits and personalities across different content topics. LLMs are typically trained on trillions of tokens, I do not see how 44k dialogues can capture the combinations of personalities and topics. In theory, this dataset also needs to be massive to cover varied domains.

4. The paper is hard to read and often unintuitive. The mathematical complexity must be simplified and replaced with more intuitive design and modeling choice explanations so that readers may grasp core ideas faster.

**Reproducibility:**

4: Could mostly reproduce the results, but there may be some variation because of sample variance or minor variations in their interpretation of the protocol or method.

**Reviewer Confidence:**

4: Quite sure. I tried to check the important points carefully. It's unlikely, though conceivable, that I missed something that should affect my ratings.

---

> ### Author Rebuttal · Authors · 2023-08-28
>
> Thank you for your valuable feedback. We would like to provide a comprehensive response addressing each of your concerns.
>
> > (1)  It is somewhat unclear if these metrics are extensively leveraged in personalized dialogue generation.
>
> We understand your reservations, classifier-based metrics have broad applications in personalized dialogue generation [1,2].
>
> > (2) limited reproducibility
>
> Thank you for highlighting this, but we’d like to clarify that our code and training scripts have been provided in the supplementary materials,  alongside detailed test dataset cases in Appendix D. We will release the further revised dataset and model checkpoint in the carema-ready version to enhance reproducibility.
>
> > (3) Comparison with Large Language Models
>
> We've conducted a comparison with ChatGPT in Table 3 and provided comprehensive inference cases, including ChatGPT results, in Appendix D.
>
> > (4)  training complexity &  practical deployment
>
> We understand your concerns. We attach a new timing table below to address training and inference cost concerns. All baselines were trained and tested on a single RTX4090, except for ChatGPT called via API.
>
> | model(GPU=a single RTX4090) | training-time(20  epoch) | inference-time(1000  items) |
> | --------------------------- | ------------------------ | --------------------------- |
> | BOB                         | 5.9hours                 | 183s                        |
> | MSP                         | 21hours                  | 730s                        |
> | CLV                         | 3.3hours                 | 429s                        |
> | LMEDR                       | 60hours                  | 142s                        |
> | **MIRACLE (Ours)**          | **3.2hours**             | **121s**                    |
> | ChatGPT (on Many A100s)     | -                        | >1000s                      |
>
> According to the attached table, we can efficiently train our model on a single RTX3090/4090 GPU for several hours. This is facilitated by disentangling complex personalities into simpler attributes. And our model also has faster inference speed compared with baselines, thanks to our flexible EBM composition and customized ODE sampling methods.
>
> > (5) insufficient dataset
>
> Dataset sufficiency is ensured for three key reasons.  (1) Our model's performance doesn't necessitate training an LLM or training from scratch, leveraging pre-trained bert-base and gpt2-medium models, as outlined in L258-265, hence doesn't demand a large volume of data.  (2) We disentangle complex personality into simpler attributes (e.g., language style, attitude, mental characteristics), efficiently aligning personality and simplifying data requirements for training. (3)  To ensure dataset quality, we've employed rigorous measures such as diverse conversation topics sampling, meticulous prompt design, multi-round human-involved cleaning, and detoxification, as outlined in L224-L236.
>
> A noteworthy observation is that the widely utilized ConvAI2 dataset, as exemplified in Table 1 of [6], is similarly sampled at a size of 43k in [6], which aligns with our dataset's sampling. The SOTA performance in Tables 1, 2 and detail cases in  Appendix D also prove the effectiveness of our dataset and methods.
>
> > (6) readability enhancement
>
> Appreciate your comments. We value specific suggestions for enhancement, though the reviewer `j4rf` praised the paper as “well-written and easily comprehensible” and `trw2` praised it as “well-motivated”. If possible, pinpointing which part is difficult to understand would greatly aid us in improving readability of the paper.
>
> > (7)  CVAE model choice
>
> CVAE is widely employed in dialogue generation [6,7,8]  due to its conditional nature, allowing it to leverage dialogue history. The structural latent space of VAE-based architecture facilitates efficient attribute modeling and manipulation[3,9]. Thus, we employ it as a foundational framework to model multiple personal factors, offering several advantages highlighted in L109-L110:
>
> - Circumvent the non-differentiable nature of the text: Direct personalization control is hard within a discrete text space due to its non-differentiable nature, which can be overcome by utilizing VAE-based latent space.
> - Better alignment/better Controllability:  Structural and compressed latent space simplifies and accelerates alignment with various attributes, ensuring better control.
>
> **reference**
>
> [1] Assigning Personality/Profile to a Chatting Machine for Coherent Conversation Generation
>
> [2] Controllable Text Generation with Focused Variation
>
> [3] Composable Text Control Operations in Latent Space with Ordinary Differential Equations
>
> [4] Quark: Controllable Text Generation with Reinforced Unlearning
>
> [5] Mix and Match: Learning-free Controllable Text Generation using Energy Language Models
>
> [6] Enhancing Personalized Dialogue Generation with Contrastive Latent Variables: Combining Sparse and Dense Persona
>
> [7] Dlvgen: A dual latent variable approach to personalized dialogue generation.
>
> [8] Learning Discourse-level Diversity for Neural Dialog Models using Conditional Variational Autoencoders
>
> [9] Controllable and Compositional Generation with Latent-Space Energy-Based Models

---

### Official Review · Reviewer_jJFK · 2023-08-11

**Typos Grammar Style And Presentation Improvements:** 1. handicraft -> handcraft
**Soundness:** 2

**Excitement:**

3: Ambivalent: It has merits (e.g., it reports state-of-the-art results, the idea is nice), but there are key weaknesses (e.g., it describes incremental work), and it can significantly benefit from another round of revision. However, I won't object to accepting it if my co-reviewers champion it.

**Paper Topic And Main Contributions:**

In this study, the researchers leverage ChatGPT to bootstrap an attributed dialogue corpus and subsequently deploy the Conditional Variational Autoencoder (CVAE) to dissect persona traits into distinct persona-related attributes systematically. The manuscript's salient contribution lies in its novel introduction of two unique losses, which, when integrated with the CVAE, promote the generation of tailored responses. Furthermore, the authors offer a dialogue corpus that has been augmented and meticulously enriched with personal attributes.

**Reasons To Accept:**

1. Provide a new corpus.
2. Detailed data pipeline and case studies.

**Reasons To Reject:**

1. Based on the data presented in Table 4, the two losses appear to have a limited impact on the overall framework.
2. The paper lacks an in-depth analysis of the Evidence Lower Bound (ELBO), a critical component for CVAE models.
3. The observed improvement in coherence is incremental.
4. Comparing the personalization metric using a classifier derived from the same dataset that also trains the generation model may raise concerns of equity in evaluation.
5. ChatGPT appears inherently equipped for this task, given that the dataset has been generated utilizing it.

**Reproducibility:**

4: Could mostly reproduce the results, but there may be some variation because of sample variance or minor variations in their interpretation of the protocol or method.

**Reviewer Confidence:**

4: Quite sure. I tried to check the important points carefully. It's unlikely, though conceivable, that I missed something that should affect my ratings.

---

> ### Author Rebuttal · Authors · 2023-08-28
>
> Thank you for your time and comments. We have carefully considered your feedback and would like to provide a comprehensive response addressing each of your concerns.
>
> > (1) The manuscript's salient contribution lies in its novel introduction of two unique losses, ...
>
> Thank you for acknowledging our unique losses. We'd like to highlight the vital role of the flexible EBM-based attribute composition in modeling complex personalities, enabling disentanglement during training and seamless composition during inference. This innovation significantly impacts performance, as evident in Table 4.
>
> To our knowledge, the EBM technique remains unexplored in the dialogue domain, and we are the first to innovatively apply it for modeling intricate multi-faceted personalities.
>
> > (2) analysis of ELBO
>
> As outlined in L285-L296 of the manuscript, ELBO comprises two key components: the response reconstruction term, ensuring generative quality conditioned on the dialogue context, and the KL divergence regularization term, fostering consistency during inference, even with previously unseen responses. We add an ablation experiment without posterior distribution to support this fundamental observation:
>
> |                     | personalization |       |       | Coherence |       |       | Fluency | Diversity |          |       |
> | ------------------- | --------------- | ----- | ----- | --------- | ----- | ----- | ------- | --------- | -------- | ----- |
> |                     | Avg.            | L.    | A.    | M.        | bleu  | rouge | nli     | ppl       | distinct | sbleu |
> | Miracle             | 92.75           | 93.30 | 93.10 | 91.86     | 45.23 | 15.21 | 70.76   | 23.68     | 70.94    | 8.90  |
> | w/o CVAE  posterior | 86.26           | 90.86 | 88.53 | 79.38     | 38.14 | 9.79  | 1.52    | 40.04     | 54.13    | 60.82 |
>
> We can observe that without the basic posterior distribution, our model experiences degradation across all metrics. Particularly, there's a catastrophic collapse observed in NLI. This is because, without guidance from $p(z|C,r)$, our encoder cannot learn the latent relationships between the response $r$ and dialogue context $C$. Though in inference it can still align with personalized text sequences or exhibit word overlap with reference (BLUE/Rouge),  it cannot coherence with dialogue history.
>
> This analysis is consistent with established VAE-based architectures [1,2], which are widely employed in dialogue domains [3,4,5,6].  For further details, we refer you to the above references.
>
> > (3) limit impact of the two losses
>
> Our proposed losses prioritize personalization metrics, which are the core of personalized dialogue agents. $\mathcal{L}_C$ improves personalization by 12.7%, while $\mathcal{L}_D$ enhances it further to 92.75 from 90.29, alongside sustained coherence, fluency, and diversity, as shown in Table 4.
>
> > (4) incremental coherence improvement
>
> While we acknowledge slight Rouge underperformance compared to LMEDR,  it is essential that our model consistently outperforms all baselines in most metrics, particularly in personalization.
>
> It is also worth noticing that, as shown in [7], Rouge has limitations (e.g. handling synonyms). Following common practices like 8,9], we use BLEU, Rouge and NLI demonstrating our approach's robustness in Coherence performance (e.g., +1.6% in BLEU compared to LMEDR) in Table 1.
>
> > (5) classifier evaluation equity concerns
>
> This is indeed a good concern, we've taken steps to address potential bias. Our approach involves utilizing a separate dataset for training the classifier, as outlined in L1164 in our paper.  To ensure alignment with human judgment, we conducted thorough human checks on the classifiers (refer to Table 7 in Appendix C.3.1).
>
> > (6) comparison with ChatGPT
>
> We recognize ChatGPT's inherent capabilities in single-attribute conversations [10], yet it faces limitations in handling multiple attributes [11, 12], as mentioned in L70-L79 of our manuscript.  In contrast, our method excels in controlling multiple personal attributes simultaneously, as evidenced by Table 3.
>
> We would also like to emphasize that our model only requires exclusive single-attribute conversations for training  (not the full personality), making it highly suitable for complex personalized response generation, as demonstrated in the table below:
>
> | model(GPU=RTX4090) | training-time(20  epoch) | inference-time(1000  item) |
> | ------------------ | ------------------------ | -------------------------- |
> | BOB                | 5.9hours                 | 183s                       |
> | MSP                | 21hours                  | 730s                       |
> | CLV                | 3.3hours                 | 429s                       |
> | LMEDR              | 60hours                  | 142s                       |
> | MIRACLE(Ours)      | **3.2hours**             | **121s**                   |
> | ChatGPT            | -                        | >1000s                     |
>
> **References**
>
> [1] Auto-Encoding Variational Bayes
>
> [2] Learning structured output representation using deep conditional generative models.[3]  Learning Discourse-level Diversity for Neural Dialog Models using Conditional Variational Autoencoders
>
> [4] Enhancing Personalized Dialogue Generation with Contrastive Latent Variables: Combining Sparse and Dense Persona
>
> [5] Fuse It More Deeply! A Variational Transformer with Layer-Wise Latent Variable Inference for Text Generation
>
> [6] Dlvgen: A dual latent variable approach to personalized dialogue generation.
>
> [7] The limits of automatic summarisation according to ROUGE.
>
> [8] Enhancing Personalized Dialogue Generation with Contrastive Latent Variables: Combining Sparse and Dense Persona
>
> [9] Improving Personality Consistency in Conversation by Persona Extending
>
> [10] Inducing anxiety in large language models increases exploration and bias
>
> [11] Red teaming chatgpt via breaking: Bias, robustness, reliability and toxicity.
>
> [12] Chatgpt an enfj, bard an istj: Empirical study on personalities of large language models.

---

### Meta-Review · Area_Chair_kqug · 2023-09-21

**Recommendation:** 3

**Metareview:**

The paper introduces a new method for personalized dialogue generation called MIRACLE, which disentangles complex personalities into three attributes (language style, attitude, and mental characteristic) and utilizes an energy-based model(EBM) for generation. The paper also proposes two new loss terms that can make personalized text generation better (aspect classification loss and attribute distance loss).

The paper has demonstrated many merits but can still be largely improved by addressing the following issues:
1. The observed improvement in coherence is incremental.
2. An in-depth analysis of the synthetic data. Do these data have sufficient diversity? Can they capture the characteristics of human dialogues well?
3. It would be nice to add the results of LLama, will the proposed approach still be much more efficient than LLama?
4. Authors are encouraged to release their code, model checkpoints, and datasets generated.

---

### Meta-Review · Senior_Area_Chairs · 2023-10-05

**Recommendation:** 3

**Metareview:**

meta review

---

### Decision · Program_Chairs · 2023-10-07

**Decision:**

Accept-Findings

**Comment:**

The paper introduces a new method for personalized dialogue generation called MIRACLE, which disentangles complex personalities into three attributes (language style, attitude, and mental characteristic) and utilizes an energy-based model(EBM) for generation. The paper also proposes two new loss terms that can make personalized text generation better (aspect classification loss and attribute distance loss).

The paper has demonstrated many merits but can still be largely improved by addressing the following issues:
1. The observed improvement in coherence is incremental.
2. An in-depth analysis of the synthetic data. Do these data have sufficient diversity? Can they capture the characteristics of human dialogues well?
3. It would be nice to add the results of LLama, will the proposed approach still be much more efficient than LLama?
4. Authors are encouraged to release their code, model checkpoints, and datasets generated.|meta review